# A Deep Convective Systems Database Derived from the Intercalibrated Meteorological Geostationary Satellite Fleet and the TOOCAN algorithm (2012-2020)

Thomas Fiolleau[1] and Rémy Roca[1]

[1]Université de Toulouse, LEGOS (CNRS/UT3), Toulouse France

*Correspondence to*: Thomas Fiolleau (thomas.fiolleau@cnrs.fr)

**Abstract.**

We introduce two databases aimed at facilitating the study of deep convective systems (DCS) and their morphological characteristics over the intertropical belt during the period spanning from 2012 to 2020: TOOCAN and CACATOES. The TOOCAN database is constructed using a tracking algorithm called TOOCAN applied on a homogenized GEOring infrared (IR) archive and enables the documentation of the morphological parameters of each DCS throughout its life cycle. The homogenized GEOring IR database has been built from level-1 data of a fleet of geostationary platforms originating from

various sources and has been inter-calibrated, spectrally adjusted, and limb darkening corrected, specifically for the high cold cloud onto a common reference, the IR channel of the ScaRaB radiometer on-board Megha-Tropiques. The resulting infrared observations are then homogeneous for Brightness Temperatures (BT) < 240 K with a standard deviation lower than 1.5 K throughout the GEOring.  A systematic uncertainty analysis is carried out. First, the radiometric errors are shown to have a little impact on the DCS characteristics and occurrences. We further evaluate the impact of missing data and demonstrate that

a maximum of 3 hours of consecutive missing images represents a favorable compromise for maintaining tracking continuity while minimizing the impact on the DCS morphological parameters. However, beyond this temporal threshold, the segmentation of DCS is significantly compromised, necessitating the interruption of the tracking process. The CACATOES database is derived from the TOOCAN database through a post-processing procedure, which involves projecting the morphological parameters of each deep convective system (DCS) onto a 1°x1°-1-day grid. This resultant dataset provides a

broader perspective, allowing for an Eulerian analysis of the DCS and facilitating comparisons with auxiliary gridded datasets on the same daily $1° \times 1°$ grid box.

Both the TOOCAN and CACATOES databases are provided on a common netCDF format that is compliant with Climate and Forecast (CF) Convention and Attribute Convention for Dataset Discovery (ACDD) standards.

A total of 15x10$^6$ DCS have been identified over the tropical regions and the 9-year period. The analysis of DCS over the
tropical oceans and continents reveals a large variety of DCS characteristics and organization. They can last from few hours
up to several days, and their cloud shield ranges from 1000 km² to a few millions of km². Oceanic DCS are characterized by a
longer lifetime duration and larger shields. Finally, the DCS geographical distribution is in line with previous DCS climatology
built from other algorithms and satellite observations.

All datasets can be accessed via the repository under the following data DOI:

- TOOCAN database: https://doi.org/10.14768/1be7fd53-8b81-416e-90d5-002b36b30cf8 (Fiolleau and Roca, 2023)
- CACATOES database: https://doi.org/10.14768/98569eea-d056-412d-9f52-73ea07b9cdca (Fiolleau and Roca, 2023)

## 1 Introduction

Deep convective systems (DCS) are central to the hydrological and energy cycle of the tropical region (Roca et al. 2014a,
Roca et al. 2010). Despite a long research history (Houze, 2018), understanding the lifetime duration of tropical convective
systems and the lengths of their various phases throughout their life cycles in the current climate, as well as their evolution in
a warmer and moister world, remains challenging (Roca et al. 2020). Deep convective systems in the Tropics can span a wide
range of scales (10-1000 km) and degree of organization (Houze 2004). Within this full spectrum of organized convection,
Mesoscale convective systems (MCS) form a specific family of deep convective systems producing contiguous precipitation
on horizontal scales larger than 100 km (Houze 2004) and which can organize into mesoscale convective complexes (Maddox,
1980), squall-line systems, super cluster (Mapes and Houze, 1993). DCS are composed of a convective core where heavy
rainfall takes place, associated to a stratiform anvil with lighter precipitation as well as nonprecipitating cirriform cloudiness.
DCS are further characterized by a life cycle. DCS initiate and develop from one or more individual deep convective cells,
which organize themselves into convective cores. The system then enters a maturity phase in which a stratiform part and
cirriform cloud expand. Finally, as convection vanishes, DCS are no longer fed by convective cells, they dissipate and scatter
into several individual cirriform clouds.

Well curated global satellite observations can provide a useful resource to constraint theoretical and modelling perspectives
on deep convective systems. In particular, after decades of field campaigns and detailed case studies, satellite climatology can
now support the statistical analysis needed to refine our current understanding. Indeed, the information related to the life cycle
of deep convective systems can only be obtained statistically at Tropical scale by using high frequency imagery available from
the geostationary orbit. Meteorological agencies around the world successfully operate geostationary satellites for more than
four decades.

From the 1980's, and in the context of campaign measurements (GATE, WMOMEX…), some automatic tracking algorithms
have been developed and implemented for the detection and tracking of convective systems from infrared (IR) imagery of

geostationary satellites. Tracking algorithms are generally based on two steps: a detection step to identify contiguous areas of
cold temperatures in a single IR image, referred to as cloud clusters in which deep convection is organized, and a tracking step
to match the identified cloud clusters from one frame to the next. A deep convective system through its life cycle then
corresponds to a succession of cloud clusters in a time sequence of IR images. Tracking algorithms have evolved over time in
line with the way to describe deep convective systems, with the improvements in geostationary imagers and with the technical
advances allowing an improvement of the identification and tracking of deep convection and their high cold cloud shields.

The detection stage is historically based on the thresholding of the IR images using brightness temperature (BT) thresholds to
delineate the cloud clusters. In the literature, a wide range of temperature thresholds from 208 K to 253 K has been used to
detect cloud clusters at different degrees of organization. Hence, Williams and Houze (1987) identified cloud clusters larger
than 5000 km² with a 213 K threshold, while Maddox (1980) applied a 241 K threshold to identify the high cold cloud shield
associated to Mesoscale Convective Complex (MCC) over the central United States. Machado et al (1992) studied the
dependence of cloud cluster size on the choice of brightness temperature threshold for different tropical regions. They found
that temperature thresholds in the range [240 K-255 K] had little effect on the surface of the cloud clusters. Over the years,
developments have been made on the detection stages to improve the characterization of cloud clusters. Thus, Mathon and
Laurent (2001) identified cloud clusters larger than 5000 km² by applying three different temperature thresholds to the infrared
imagery: a 213 K temperature threshold to discriminate deep convective cores, a 233K threshold commonly used to estimate
surface precipitation in the tropics (Arkin, 1979) and a 253 K threshold to identify the boundaries of the high cold cloud shield.
In this way, it is possible to access the number of convective cores included in a high cloud shield. This technique has been
adopted by Evans and Shemo (1996) and Ocasio et al. (2020) with some slightly different brightness temperature thresholds.

However, detection step based on single brightness temperature thresholds face some difficulties to catch the complexity of
the full spectrum of convective organization (Fiolleau and Roca, 2013a). A single BT threshold tends to identify cloud cover,
but cannot differentiate the components that make up convective systems. To improve the high cold cloud segmentation, Boer
and Ramanathan (1997) have developed an algorithm called Detect And Spread (DAS). It relies on the assumption that pixels
adjacent to a convective core in an IR satellite image belong to the same physical cloud cluster, and that the optical depth of
cloud cover decreases from the convective core to the edge of the cloud cluster. The algorithm is based on a region growing
technique consisting in the application of multi brightness temperature thresholds on the IR images to identify the convective
cores which are then spread to reach the boundaries of the cloud shield. Similar approaches have been adopted by Roca and
Ramanathan (2000), Feng et al. (2021) , Heikenfeld et al. (2019) , Wilcox (2003),  Rajagopal et al. (2023), Autones and
Moisselin (2013).

Regarding the tracking step, Williams and Houze (1987) introduced the area-overlapping technique to link cloud clusters in
successive images, building the convective systems from their initiation to their dissipations. This technique has been widely
used for years (Arnaud et al. 1992; Mathon and Laurent 2001 ;Machado et al. 1998), and has been the subject of numerous

technical evolutions to increase the matching accuracy between clusters identified in two successive time steps. Hence, to enhance the cloud matching, the projection of the cloud position has been added in some tracking methods (Ocasio et al. 2020, Feng et al. 2023). Huang et al. (2018) combined a Kalman filter technique with the area-overlapping method to improve the tracking of small and fast-moving convective systems. Some algorithms have based their tracking stages on a search radius method and on the prediction of the position of the cloud cluster's center of mass to link cloud clusters to each other (Heikenfeld et al. 2019, Woodley et al. 1980, Ostlund 1974). Carvalho and Jones (2001)  and Endlich and Wolf (1981) used the maximum spatial correlation between cloud clusters of successive images, when Hodges (1994) has developed a minimization of cost function to improve the cloud cluster correspondence issue between two-time steps.

However, tracking algorithms based on the two-step technique (detection step and tracking step) can result in artificial splitting and merging artefacts throughout DCS life cycles that require specific post-processing (Feng et al. 2021, Williams and Houze 1987, Mathon and Laurent 2001), and which remain a major source of uncertainty when considering DCS life cycle analysis (Prein et al., 2024).

To overcome these issues, another branch of algorithms has then been developed based on the assumption that convective systems can only be tracked if they are considered as 3-dimensional (space+time) objects in their space-time domain. These algorithms do not work anymore IR image per IR image, but process a volume of IR images in 3 dimensions (longitude, latitude, time), to identify and track convective systems in a single step. Some algorithms then apply a single brightness temperature threshold in the spatio-temporal domain to detect and track deep convective systems (Mapes et al. 2009, Dias et al. 2012, Prein et al. 2024 , Poujol et al. 2020). Other tracking algorithms use a more complex technique and apply a 3D multi-step, multi-thresholding technique, derived from the detect and spread method (Boer and Ramanathan, 1997), on the volume of IR imagery (Fiolleau and Roca 2013a, Jones et al. 2023). With such a 3D segmentation technique, high cold cloudiness is decomposed in deep convective systems which are not impacted by splitting and merging events as for previous tracking algorithms. At the beginning of the DCS life cycle, the individual convective cells which develop, are part of a same DCS, and feed its anvil. Similarly, at the end of DCS life cycle, the scattered cirriform clouds all belong to the same DCS which produced them.

These tracking algorithms have given rise to a large corpus of literature at the regional scale but only a few deep convective systems databases have been made available at global scale so far based on various global infrared archive (Table 1). These climatologies provided an initial perspective at tropical scale, revealing the ubiquity of mesoscale systems with various durations and spatial extents across a wide spectrum of large-scale environments. This insight has prompted numerous scientific investigations. For instance, Elsaesser et al. (2022) investigated the growth rate of the cloud shield across the entire tropical belt. Using the CLAUS dataset, the statistics of DCS triggering revealed the strong link to orography while spectral analysis brought observational support to tropical dynamics theory (Dias et al., 2012). The overwhelming contribution of the long-lasting systems to the tropical precipitation total amount was quantitatively estimated (Roca et al., 2014) as well as to the

extreme precipitation (Roca and Fiolleau, 2020). The recent decades tropical trends in precipitation has been shown to be associated with the trends in the occurrence of deep organized systems (Tan et al., 2015). These examples are strong incentive

to build and sustain IR derived database of the DCS morphology in support of climatological investigations of the tropical water cycle (Feng et al., 2021).

| IR source | External source | Period | Coverage | resolution | | Homogenization across the fleet | | | | | algorithm technique | | Reference |
|---|---|---|---|---|---|---|---|---|---|---|---|---|---|
| | | | | Space | Time | Spatial res. | Temp res. | Spectral | Inter-calibration | VZA correction | Detection technique | Tracking technique | |
| *ISSCP-B3* | *No* | *1983-2009* | *55°S-55°N* | *30km* | *3h* | *yes* | *yes* | *No* | *yes* | *no* | *Single BT threshold* | *area-overlapping* | *(Machado and Rossow, 1993) (Vant-Hull et al., 2016)* |
| ISCCP-B3 | No | 1989-1993 | 40°S-40°N | 30km | 3h | Yes | Yes | No | Yes | no | Single BT threshold | area-overlapping | (Tsakraklides and Evans, 2003) |
| *CLAUS* | *No* | *1985-2008* | *30°S-30°N* | *30km* | *3h* | *yes* | *yes* | *No* | *yes* | *yes* | *Single BT threshold* | *Kalman filter combined with area-overlapping* | *(Huang et al., 2018)* |
| CLAUS | No | 1985-2008 | 30°S-30°N | 30km | 3h | yes | yes | No | yes | yes | 3D clustering technique | area-overlapping | (Dias et al., 2012) |
| Gridsat | IBTrACS | 1980-2008 | Tropical oceanic basins | 8km | 3h | Yes | Yes | No | Yes | | Single BT threshold | area-overlapping | (Hennon et al., 2013) |
| CPC Mosaic | No | 2002-2012 | 60°S-60°N | 4km | 30' | Yes | Yes | No | No | Yes | Single BT threshold | area-overlapping | (Esmaili et al., 2016) |
| GEO Raw data | No | | 60°S-60°N | | | No | No | No | No | No | Single BT threshold | area-overlapping | (Laing and Fritsch 1997) |
| GEOring Raw data | No | Summer 2012-2014 | 30°S-30°N | 2 to 5km | 30' | No | Yes | No | None | None | 3D Detect And Spread (TOOCAN) | | (Roca et al. 2017) |
| *CPC Mosaic* | *iMERG IBTrACS* | *2001-2019* | *60°S-60°N* | *0.1°* | *1h* | *yes* | *yes* | *No* | *None* | *Yes* | *Detect And Spread* | *area-overlapping* | *(Feng et al., 2021)* |
| GEOring Raw data | No | 2012-2016 | 30°S-30°N | 0.04° | 30' | Yes | Yes | Yes | Yes | Yes | 3D Detect And Spread (TOOCAN) | | (Bouniol et al., 2021) (Elsaesser et al., 2022) |

**Table 1** Technical characteristics of the existing DCS databases over the entire tropical band. The DCS databases made available and associated with a DOI are indicated in *italic*

On the other hand, the limitations of the IR information to document the precipitating processes occurring at the sub storm

scale are well known (Liu et al., 2007) yet the IR based convective cloud shield analysis can shed significant insights into the dynamics of the system. A simple two stages model can well describe the life cycle of the DCS shield and that the morphological parameters of the storms are so tightly related that eventually the full life cycle of the cloud shield can be reconstructed knowing for instance only the duration and the maximum extension of the system (Roca et al., 2017). IR derived information about the DCS life are crucial to contextualize overpassing observations, be it active or passive microwave

observations (Fiolleau and Roca 2013b, Bouniol et al. 2016) ). The knowledge of the cloud shield can further inform physically based precipitation retrievals (Bellerby et al. 2009, Guilloteau and Foufoula-Georgiou 2024). Such variety of applications is one more strong motivation to elaborate upon the infrared archive to document the life cycle of DCS in support of process studies.

Cloud tracking algorithm further add requirements to the GEOring[1] dataset used to elaborate global MCS climatology. While a few studies rely on 3 hourly data, it is important to make use of at least 30 minutes imagery to describe the full spectrum of deep convective systems (Fiolleau and Roca, 2013a). Similarly, high resolution observation is required to address the full life cycle of the convective systems that could otherwise be truncated (Schröder et al., 2009). The variety of spatial and spectral resolution for instance for the period 2012-2020 (see discussion below) further calls for a homogenization effort at the level 1 prior to running the tracking algorithm (Fiolleau et al., 2020).

This paper presents such a DCS database built from a homogenized IR archive and a tracking algorithm called TOOCAN (Tracking Of Organized Convection Algorithm using a 3-dimensional segmentatioN) (Fiolleau and Roca, 2013a). In Section 2, we present the harmonized infrared geostationary dataset, and the way we have performed the homogenization procedure using the ScaRaB-Megha-tropiques IR observation. We will then introduce the IBTrACS dataset, which will be used to filter out DCS belonging to a cyclonic circulation or if these DCS are classified themselves as cyclonic events. The functioning of the TOOCAN algorithm is introduced section 3. We will discuss the possible uncertainty on the deep convective systems occurrences and characteristics induced by the residual error of the homogenization procedure, as well as the geostationary data availability. Exemplary illustrations of the database are shown in section 4 and finally, the data availability and format are further discussed in Section 5.

## 2 Data

### 2.1 The harmonized infrared geostationary dataset 2012-2020

Thermal channel brightness temperature data obtained by the operational meteorological geostationary satellite fleet are used to monitor the deep convective systems over the tropical belt for the whole 2012-2020 period. Table 2 shows the technical characteristics of the fleet of geostationary platforms (GEOring) and their associated IR channels over the 9-year period. As shown in Fiolleau et al. (2020), the GEOring is far from a homogeneous suite of instruments operated in similar fashion. Spatial and temporal resolutions, as well as the spectral filter functions and the calibration procedures differ from one platform to another. All of these technical differences lead to biases in the brightness temperature measurement from one geostationary platform to another one. To overcome such issues, Fiolleau et al. (2020) have then developed a methodology of homogenization of the thermal infrared channels of the meteorological geostationary satellite fleet for cold cloud studies and based on the IR channel of the Scanner for Radiation Budget (ScaRaB) onboard Megha-Tropiques. This homogenization procedure includes the computation of the inter-calibration and spectral normalization coefficients every 10 days as well as the correction of the limb darkening effects impacting the brightness temperature measurements in the range [180K-240K] from geostationary satellites for the 2012-2016 period. The extension of the geostationary database until 2020 has required to continue this

---

[1] The GEOring refers to the whole geostationary platforms which data all together covers all the longitude of the Earth surface.

harmonization effort. These four additional years are characterized by the launch of new geostationary satellites. Over the
Indian ocean, METEOSAT-7 has been replaced by MSG-1 in January 2017, the new generation GOES-16 has replaced GOES-
13 over Americas in December 2017. Finally, over the Eastern Pacific region, the GOES-15 platform has been shifted from
its initial nadir at 135° W to 128° W in November 2017 to ensure continuity of observations in the Eastern Pacific before being
replaced by GOES-17 in December 2019. From the end of 2018, the Megha-Tropiques satellite suffers from technical issues
on the data management subsystem, implying a decreasing of the availability of the ScaRaB data. From this date, only about
30% of the data are available. Nevertheless, the quality of the ScaRaB measurements is not impacting by such technical issues,
and have the same performance as at the beginning of the mission. To overcome the lack of ScaRaB observation, the calculation
of inter-calibration and spectral normalization coefficients was carried out every 30 days instead of the initial 10 days.

| Region | Platform | Nadir location | Instrument | Central wavelength | Spectral interval | Spatial resolution at nadir | Temporal resolution | Spatial coverage | Source | Period |
|---|---|---|---|---|---|---|---|---|---|---|
| EASTERNPACIFIC | GOES-15 | 135°W | IMAGER | 10.7 µm | 10.2-11.2 µm | 4 km | 30 min | 180°W-105°W 55°S-55°N | NOAA/DWD | 01/2012-11/2017 |
| EASTERNPACIFIC | GOES-15 | 128°W | IMAGER | 10.7 µm | 10.2-11.2 µm | 4 km | 30 min | 180°W-105°W 55°S-55°N | NOAA/DWD | 11/2017-12/2019 |
| EASTERNPACIFIC | GOES-17 | 137°W | IMAGER | 11.2 µm | 10.4-12.0 µm | 2 km | 10 min | 170°W-80°W 55°S-55°N | AERIS/NOAA | 12/2019-12/2020 |
| AMERICA | GOES-13 | 75°W | IMAGER | 10.7 µm | 10.2-11.2 µm | 4 km | 30 min | 111°W-30°W 55°S-55°N | NOAA/DWD | 01/2012-12/2017 |
| AMERICA | GOES-16 | 75°W | IMAGER | 11.2 µm | 10.4-12.0 µm | 2 km | 10 min | 130°W-20°W 55°S-55°N | AERIS/NOAA | 12/2017-12/2020 |
| AFRICA | METEOSAT-8/9/10/11 | 0° | SEVIRI | 10.8 µm | 9.8-11.8 µm | 3 km | 15 min | 45°W-45°E 55°S-55°N | EUMETSAT/ AERIS | 01/2012-12/2020 |
| INDIA | METEOSAT-7 (IODC) | 57.5°E | MVIRI | 11.5µm | 10.5-12.5 µm | 5 km | 30 min | 30°E-107°E 55°S-55°N | EUMETSAT/ AERIS | 01/2012-01/2017 |
| INDIA | METEOSAT-8 (IODC) | 40.5° | SEVIRI | 10.8 µm | 9.8-11.8 µm | 3 km | 15 min | 30°E-95°E 55°S-55°N | EUMETSAT/ AERIS | 01/2017-12/2020 |
| WESTERNPACIFIC | MTSAT-2 | 145°E | IMAGER | 10.8 µm | 10.3-11.3 µm | 4 km | 30 min | 94°E-170°W 55°S-55°N | AERIS/ CIMSS | 01/2012-05/2015 |
| WESTERNPACIFIC | HIMAWARI-8 | 140.7°E | AHI | 11.2 µm | 11.0-11.4 µm | 2 km | 10 min | 94°E-170°W 55°S-55°N | AERIS/ JMA | 06/2015-12/2020 |

**Table 2** Technical characteristics of the operational geostationary satellites fleet and the associated imagers used over the
2012-2020 period.

Figure 1a shows the time series of the initial bias in BT for all the geostationary imagers with respect to the ScaRaB observation
in the range [180 K-240 K], and for geostationary zenith angle lower than 20° over the 2012-2020 period. The large bias
observed in the MET7/MVIRI calibration between 2012 and 2017 (Figure 1a) is explained by significant ice contamination
on the MVIRI optics, as fully discussed in Hewison et al. (2013) and Fiolleau et al. (2020). The results of the inter-calibration
and spectral normalization is shown Fig. 1b. Over the entire period and for all the geostationary platform, the decadal residual
bias indicates a very small and stable bias (< 0.04 K) with a standard deviation lower than 0.07 K between $BT_{SCARAB}$ and the
inter-calibrated $BT_{GEO}$ ($BT_{GEO}$*). The decrease of the ScaRaB data availability does not impact the quality of the corrections
from the end of 2018. The limb darkening correction is the final step of the harmonization process and is performed
geostationary satellite per geostationary satellite. The method has been fully described in Fiolleau et al. (2020). Figure 2 shows

the variation of the biases of BT$_{GEO}$ for values lower than 235K between pairs of geostationary satellites monitoring a same region according to the difference of their zenith angles (ΔVZA), before and after zenith angle corrections. Without any corrections, BT$_{GEO}$* bias varies from -5 K to +5 K as the ΔVZA moves from -50° to 50° regardless the geostationary platform (Figure 2a). By applying the zenith angle correction, this bias averages 0.21 K with a standard deviation of 1.33 K throughout the GEOring, independent of the variation in ΔVZA.

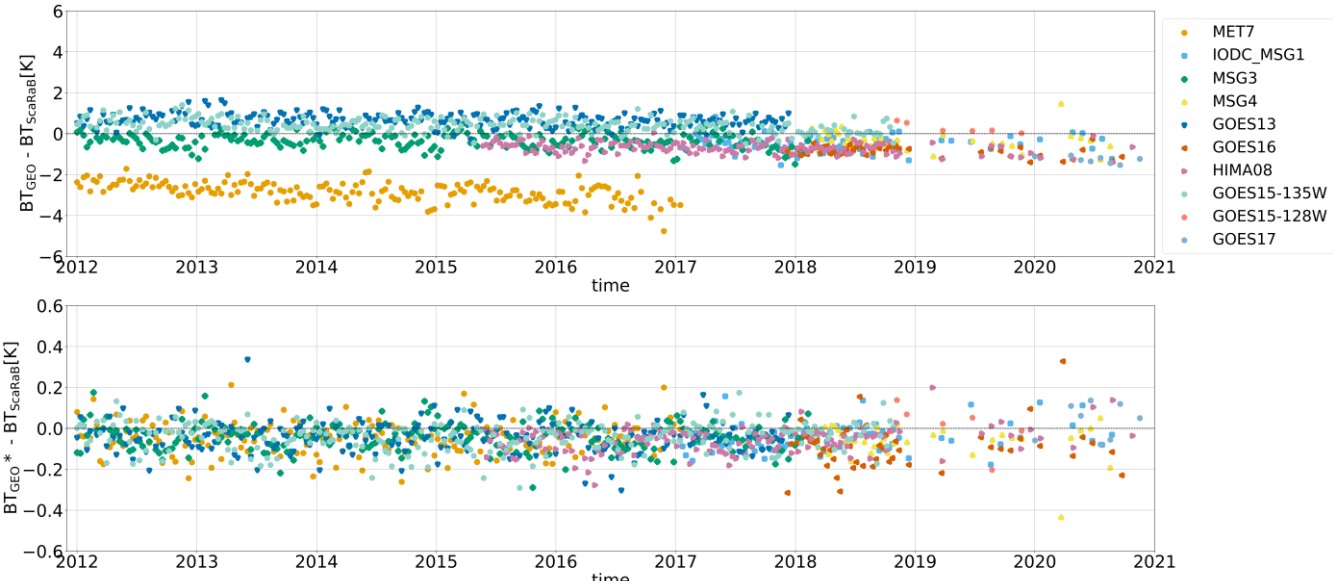

**Figure 1 (a)** Time series of initial BT bias of geostationary IR observations with respect to SCARAB in the range [180 K-240 K] between 2012 and 2020; **(b)** after spectral and calibration corrections.

To finalize the harmonization procedure, the temporal resolution has been unified to 30 minutes across the GEOring and all the geostationary data have been remapped into a common longitude-latitude 0.04° equal-angle grid (Fiolleau et al., 2020). The spatial coverage of each geostationary has been chosen relatively wide in longitude to have an important overlapping area between adjacent geostationary platforms, and between 55° S and 55° N in latitude.

GOES-13 and GOES-15 sensors follow a complex scanning sectors schedule. The full disk images are produced every 3 h, while the north hemisphere and the south hemisphere images are produced every 30 minutes with a time lag of few minutes between each scan. Thus, to get a full disk image at a 30-minute temporal resolution, the southern and northern scans of GOES-13 and GOES-15 have been concatenated. The acquisition scheme of the MTSAT-1 and 2 platform (from January 2012 to May 2015) over the Western Pacific region only provides northern hemisphere imagery at a 30-minute frequency; the southern hemisphere zone is only available every hour. Similarly, the southern hemisphere of a small region between 118° W and 108° W is only monitored at a 3 h temporal resolution by GOES-15.

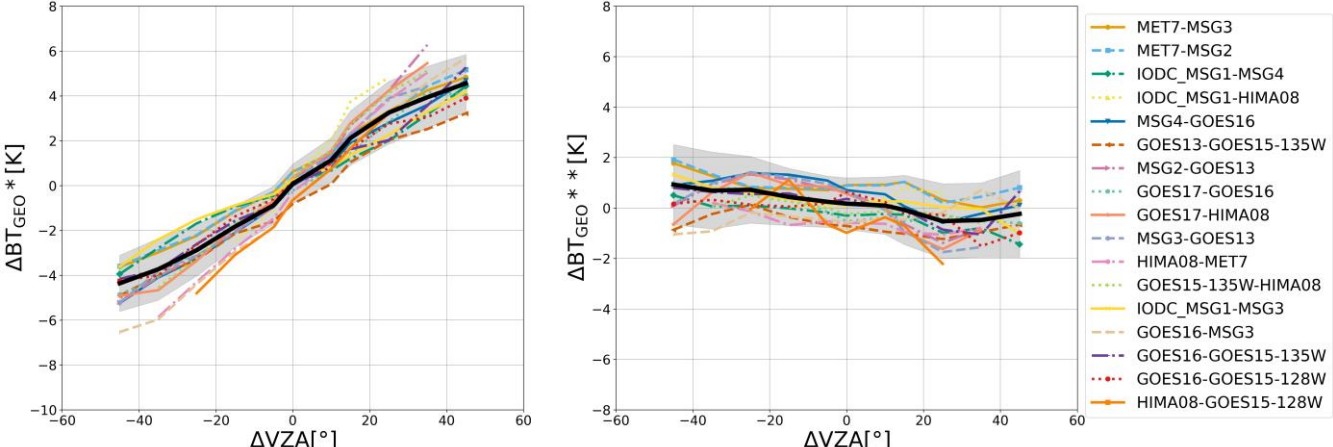

**Figure 2 (a)** Variation of the BT bias according to the VZA differences between pairs of geostationary platforms observing common areas and for $BT_{SCARAB}$ in the range [180 K-235 K] between 2012 and 2020 before Viewing Zenith Angle corrections; **(b)** after Viewing Zenith Angle corrections. The BT bias and its standard deviation for all the pairs of geostationary platforms are represented respectively by the black line and the filled area in grey.

Low-quality infrared images have been filtered out by applying a quality control on all the geostationary IR data (Szantai et al., 2011). Overall, the availability of all the geostationary platforms during the 2012-2020 period exceeds 96.8 % with the highest availability over Africa (99.6 %) and the lowest availability for the Western Pacific region (95 %). Figure 3 shows the time series of the missing data for all the geostationary platforms. METEOSAT-7 over India is impacted during the boreal summer by solar eclipses and data are not disseminated during a couple of hours each day from early August to mid-September and from February to March. We can notice that the replacement of METEOSAT-7 by MSG-1 in January 2017 increases drastically the data availability. Similarly, the time series highlights the improvement of data availability as the configuration of the fleet changes over time. Thus, the rate of missing data falls from 9.6 % to 1.77 % with the operationalization of HIMAWARI-8 over Western Pacific from June 2015 on-ward. The replacement of GOES-13 by GOES-16 over the American region in January 2018 improves also remarkably the data availability from 93.8 % to 98.6 %. It is also to be noticed that the new generation of the GOES platforms is not anymore impacted by rapid scan operations as was the case with older platforms, which prevented the scanning of the southern hemisphere of the American and Eastern Pacific regions. Such operation modes (orange curves) impact a high number of days during the coverage period of GOES-13 (13 % of the days) and GOES-15 (8.76 % of the days). The deployments of GOES-16 in 2018 and GOES-17 in December 2019 to their respective nadir positions over respectively the American and Eastern Pacific regions highly improve the observation of the southern regions. The overall

availability of the GEOring data over the entire period is then well suited to document the deep cloudiness in the tropical regions.

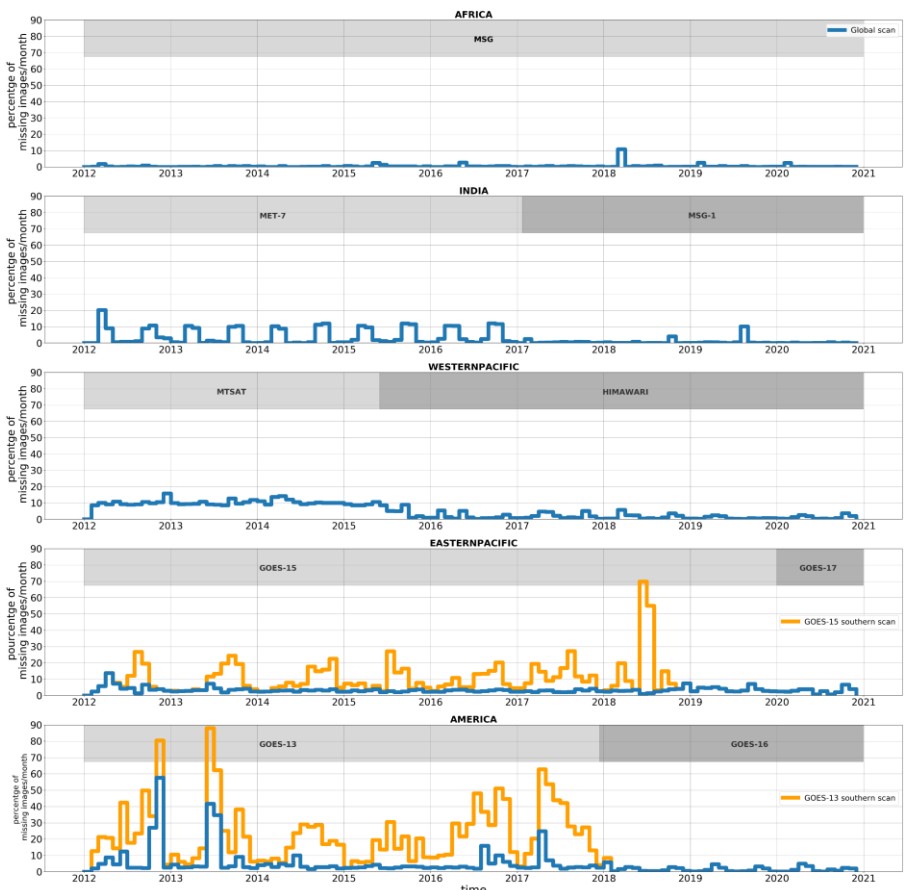

**Figure 3** Time series of the percentage of missing IR data per month for each region of interest and geostationary platform over the 2012-2020 period. Orange lines correspond to the availability of the southern scans of GOES-13 and GOES-15 which can be impacting by Rapid Scan Operation modes.

## 2.2 IBTrACS dataset

To understand the general characteristics and behavior of tropical DCS, it is important to filter out the cloud systems affected by atmospheric conditions, associated with tropical cyclones. To this end, we use the IBTrACS dataset to flag DCS belonging to a cyclonic circulation or classified themselves as tropical cyclones, allowing to either filter them out of the analysis or keep them as needed.

The International Best Track Archive for Climate Stewardship (IBTrACS) version 4 is a dataset combining all of the tropical cyclone best-track data from all the World Meteorological Organization (WMO) Regional Specialized Meteorological Centers

(RSMCs), as well as other national agencies (e.g., the Joint TyphoonWarning Center) over the globe (Knapp et al., 2011). The dataset contains the tropical cyclone location, maximum sustained wind (MSW) and minimum central pressure (MSP) as well as additional information depending on the forecast center every 6h along the lifetime of the tropical cyclone. The tropical storm positions are interpolated in time to 3 hourly positions. In this study, we will determine whether a deep convective system is closed to a tropical cyclones meteorological event at 00h00 UTC, 06h00 UTC, 12h00 UTC, and 18h00 UTC. The

combination procedure between convection systems and tropical storms is described in section 3.3. The WMO standard for MSW is a 10-min average (WMO 1983), which is used at many of the forecast centers. However, some forecast agencies use a different temporal average for MSW. A 1-min average for MSW is used at the United States forecast centers (JTWC, NHC, and CPHC), a 2-min average at the Chinese Meteorological Administration's Shanghai Typhoon Institute, and a 3-min average used at the India Meteorological Department. A conversion of all MSW to a common temporal average is then required to

analyze statistically tropical cyclone over the tropics and several conversion factors have already been determined by Harper et al. (2008). For our study, we will use the same factor 0.88 than the one used in (Kruk et al., 2010) to convert all the basins' MSW to a 1-min average. The MSW is then used to classify all the tropical cyclones according to the Saffir-Simpson hurricane wind scale (SSHS). Over the 2012-2020 period, 825 meteorological events are classified as tropical storms. Among them, 410 tropical cyclones have been identified. 208 of them have reached the SSHS category 3 (90 kt) during their lifetime, 142 the

SSHS category 4, and 17 the SSHS category 5 (136 kt).

## 3 TOOCAN algorithm

### 3.1 The principle

The functioning of the TOOCAN (Tracking Of Organized Convection Algorithm through a 3-D segmentation) algorithm has been fully described and explained in Fiolleau and Roca (2013a). The algorithm relies on a conceptual model of a convective

system consisting in an initiation phase in which deep convective cells develop and organize in a convective core, a maturity phase in which an anvil cloud develops associated to its convective core followed by a dissipation stage, in which no more convection occurs and the system breaks up into multiple cirriform clouds. Both in the spatial and temporal domains, the optical depth of cloud cover decreases from the convective core to the edge of the anvil cloud, as the brightness temperature increases. This conceptual model of a convective system corresponds then to a 3D (longitude, latitude, time) cloud cluster

made up of a convective core associated to its stratiform anvil and cirriform clouds evolving in the space-time domain.

To identify such a 3D-cloud cluster, the algorithm works within a time sequence of IR images, and applies a 3-D region growing technique, to decompose the high cold cloud shield, defined by a 235 K threshold in the spatio-temporal domain into component DCSs. This technique consists in an iterative process of detection and dilation of convective seeds in the spatio-temporal domain to detect and track DCS in a single 3D segmentation step. Individual convective seeds are first detected by

270 applying a cold brightness temperature threshold set at 190 K on the volume of IR images. If any, convective seeds with a minimum lifetime duration of 3 frames (1h30) and exceeding 625 km² per frame are kept and are spread in the spatio-temporal

domain until reaching the intermediate cold cloud shield boundaries identified at a 2K warmer BT threshold. This dilation step involves adding pixels belonging to the intermediate cold cloud shield to all previously detected convective seeds using a 10-connected spatiotemporal neighborhood kernel operator, composed of an 8-connected spatial neighborhood and a 2-connected temporal neighborhood, to favor spatial spread over temporal spread. The pixel aggregation process is constrained by a brightness temperature difference between the edge pixels and the already identified pixel, which has to be greater than -1 K to minimize the effects of local minima. Then, a new detection is applied at the 192 K threshold to detect the convective seeds too warm to be identified at the previous Tb threshold (190 K). If any, all the convective seeds are then spread until reaching the intermediate cold cloud shield boundaries at a 2 K warmer BT threshold (194 K). This iterative process of detection and dilatation is repeated with a 2K detection step from 190 K to 235 K and is stopped when all the pixels below 235K are associated with a DCS. The very cold 190 K threshold is required to identify very deep convective cores which occur in the Tropics. The multi-BT thresholds between 190 K and 235 K allows to identify the wide variety of convective cores which occur and may be more or less deep. Thanks to its spatio-temporal region-growing technique, the TOOCAN algorithm can track DCS by suppressing split and merge artifacts throughout their life cycles, which are inherent to classic overlap-based tracking techniques.

The first spatio-temporal volume of IR images is built by accumulating 15 days of geostationary infrared data in which TOOCAN operates. TOOCAN is applied in the next 15-day of volume images with a sliced-window technique, allowing a continuity of the tracked DCS between the 2 successive spatio-temporal volumes. With regard to big data processing, the algorithm can face missing IR data. When low-quality or missing images are encountered in the time series, the data available at previous and subsequent time steps are replicated, so that the tracking is carried out nominally, enabling continuity of the deep convective cloud life cycle. However, beyond a given number of successive missing images, the TOOCAN process has to be stopped, and a fresh start has to be operated at the end of the interruption. DCS impacted by this interruption are all terminated and new systems are considered to initiate at the end of the interruption, causing artificial life cycles and some biases in lifetime duration distributions. Section 3.2b will detail the sensitivity of the DCS characteristics to the data availability.

Figure 4a and 4b show a time series of the TOOCAN segmented images from the HIMAWARI-8 IR data between 12[th] October 2015 at 0h00 UTC and 13[th] October 2015 18h00 UTC over the Western Pacific Ocean. The high cold cloud shield defined by a 235 K threshold on the infrared imagery is decomposed into several deep convective systems whose anvil clouds touch each other. The full spectrum of deep convective systems organization is identified ranging from small, short-lived and isolated systems to long lasting systems, reaching several thousands of kilometers per square meters and propagating over several hundred kilometers. For instance, the deep convective systems A and C identified over the Pacific Ocean on the time series last 60 h and 137 h respectively, reach a cold cloud surface larger than $4.5 \times 10^5$ km², and propagate on a distance greater than 2000 km. The convective system B, has a lifetime duration of 22.5 h with a $2.5 \times 10^5$ km² maximum extent and propagates Westward over 890 km. All these very large and long-lived convective systems belong and contribute to complex convective

situations, sharing common high cloud cover with various convective systems that exhibit a wide range of morphological characteristics, with which they interact. Other convective systems are more isolated during their life cycles. This is the case of the DCS D and E over the Philippines islands which last ~4 h and reach a maximum cold cloud surface of ~5500 km². These systems are also characterized by their relatively stationarity, and exhibit a propagating distance of 123 km and 74 km, for DCS D and E respectively.

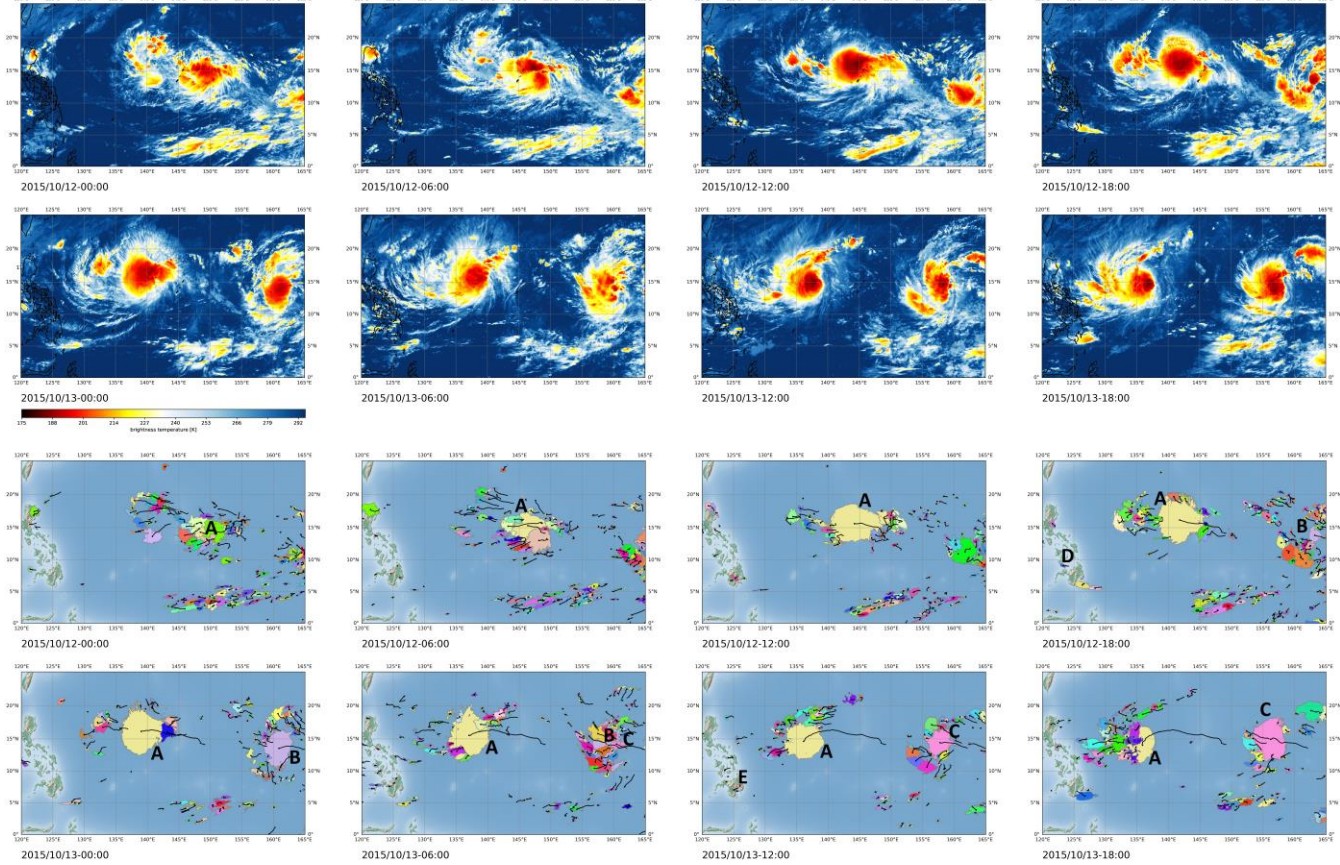

**Figure 4** Illustration of the TOOCAN segmentations for a convective situation which occurred in October 2015, over the Western Pacific region, which is also presented as Video S1 in the supplement (Fiolleau, 2024). (a) Infrared observation of HIMAWARI-8 every 6h from 12 October 2015 at 00:00 UTC to 13 October 2015 at 18:00 UTC; (b) DCS segmented by the TOOCAN algorithm. Each color corresponds to a unique deep convective system. The black lines indicate the trajectories of their centers of mass since initiation.

**3.2 Uncertainty estimation**

In this section, we will focus on assessing the impact of radiometric errors and missing images on the performance of the segmentation and tracking of deep convective systems from IR imagery as well as on the error propagation in statistical analyses of DCS.

**3.2.a Uncertainty estimation due to radiometric errors**

As discussed in section 2.1, IR observation have been homogenized with an error lower than 1.5 K throughout the GEOring considering the inter-calibration, spectral normalization and limb darkening corrections. In the following, we assess the impact of such a residual error on the morphological characteristics and occurrences of the DCS. For that, the analysis is based on the MSG IR dataset over the June to September 2012 period over the 40° W-40° E; 30° S-30° N region.

From this reference dataset, we have produced an ensemble of 134 MSG-1 IR datasets over the same region and period, whose brightness temperatures have been biased by a value estimated from a gaussian distribution with a 1.5 K standard deviation (Figure 5). The TOOCAN algorithm has been applied on these 134 biased MSG IR datasets in order to produce an ensemble of different DCS segmentations. The average distribution of the DCS lifetime durations computed from the ensemble of MSG IR dataset (blue line) shown in Fig. 6 indicates that maximum of population occurs for systems lasting in the range 0-5 h with around $3x10^4$ DCS. Then, the DCS population decreases as the lifetime duration increases. In average, around 320 DCS last more than 20 h on the 134 runs of TOOCAN.

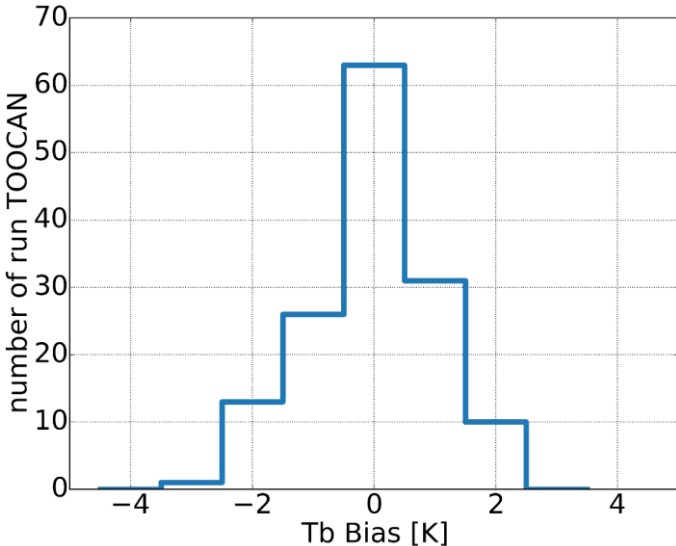

**Figure 5** Distribution of the brightness temperature bias applied on the MSG dataset between June to September 2012 and which has been built from a gaussian distribution with a 1.5K standard deviation.

The relative confidence interval (orange line) is shown Fig. 7 as a function of the lifetime duration and is computed as the absolute 95 % confidence interval divided by the DCS averaged occurrence multiplied by 100 %.

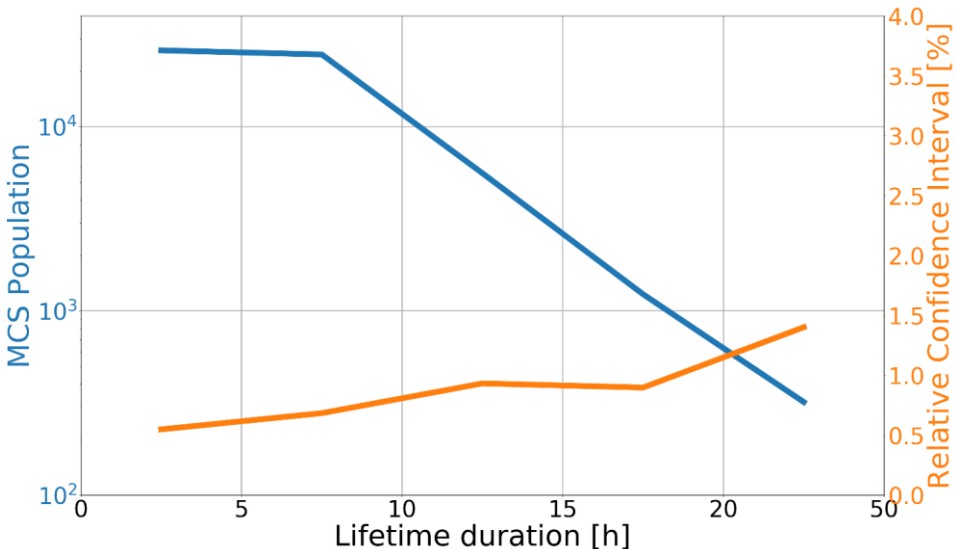

**Figure 6** Distribution of the average DCS lifetime duration (blue curve) computed from the 134 perturbated runs of TOOCAN applied on IR imagery of MSG from June to September in 2012 over the 40°W-40°E;30°S-30°N region and the associated
relative confidence interval (orange).

This sensitivity study reveals a small relative confidence interval whatever the bins of lifetime duration. For systems lasting less than 5h, a 0.54 % relative confidence interval is observed for a population of ~26000 DCS, meaning that we are 95 % confident that the true DCS population < 5 h is between 25859 and 26240 with a 1.5 K residual error. This relative confidence interval increases slightly as the lifetime duration increases and DCS population decreases. However, the relative confidence
interval remains relatively weak with a value ~1.40 % for DCS lasting more than 20 h. For these long-lasting systems, the absolute confidence interval of the DCS population is between 315 and 320.

Given the assumed 1.5 K residual bias throughout the GEOring, this analysis has shown that the IR segmentation and the DCS tracking by the TOOCAN algorithm as well as the resulting DCS lifetime duration distributions are not sensitive to such an error source.

**3.2.b Uncertainty estimation due to the IR geostationary data availability**

The sensitivity of the DCS morphological parameters is now evaluated according to the availability of the IR geostationary images. From the point of view of cloud tracking, the major problem lies less in isolated missing data than in the number of successive missing images over time, which have an impact on the continuity and quality of cloud tracking. As seen previously in section 3.1, if any missing images are found in the time series, they are replaced by the data available in the previous and

355 following time steps, allowing convective systems to be tracked. It is then important to assess the impact of consecutive missing images on the statistics of the DCS occurrence, and on the characterization of their morphological parameters.

Several scenarios arise for the DCS facing such a time period of consecutive missing images. DCS that were supposed to initiate during this missing data gap are either detected at the time of resumption, in which case their duration is artificially shortened, or they cannot be identified at all. DCS that were supposed to dissipate during this period of missing images could

dissipate at the time of resumption, in which case their lifetime duration is artificially extended. There is the case of DCS that started before the series of missing images and are expected to persist thereafter can still be tracked despite this interruption. Some DCS cannot survive the period of missing data and then dissipate artificially. Smaller systems may not survive this interruption, reducing their lifetime, or their lifetime may be artificially extended due to data replication.

Also, this sensitivity study will help us to determine the threshold on the number of successive missing images beyond which

the DCS parameters are too degraded, so that the TOOCAN process has to be interrupted.

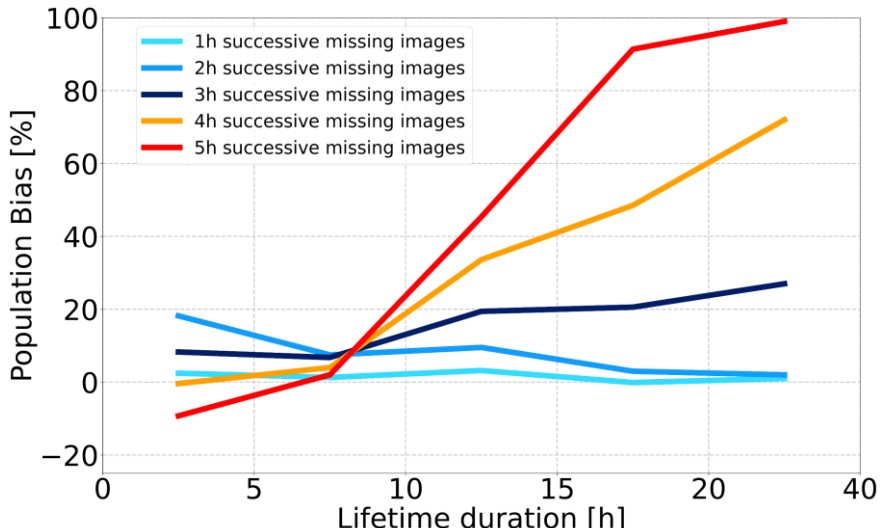

**Figure 7** Bias on the DCS population according to the lifetime duration between a reference run of TOOCAN applied on a complete MSG IR dataset in June-September 2012 over the 40°W-40°E; 30°S-30°N region and five runs of TOOCAN applied on similar MSG IR dataset but degraded every day with 1h, 2h, 3h, 4h and 5h successive missing images.

The same MSG IR observations over the African and Atlantic region from June to September 2012 constitute our baseline for

this analysis. This database is characterized by no missing data over the study period (Figure 3). To better understand the impact of the missing data periods on the DCS parameters, the TOOCAN algorithm is first applied on this reference MSG IR database, and then is applied on this same database but degraded by deleting everyday 1h of consecutive images (between 19:00 UTC and 20:00 UTC), 2h (between 19h00 UTC and 21h00 UTC), 3h (between 19h00 UTC and 22h00 UTC), 4h (between 19h00 UTC and 23h00 UTC) and finally 5h (between 19h00 UTC and 00h00 UTC) of consecutive images. The

missing data periods are filled by duplicating the available MSG IR data just before and after the missing data gap.

We have focused our analysis on the late hours of each day to mimic the MFG eclipse seasons, during which consecutive missing data occur in the evening. A gap of successive missing data in the afternoon would be more impactful on the DCS population than at night, due to the diurnal cycle of initiation. However, our aim here is to determine the duration of consecutive missing data beyond which tracking the longest-lived and largest DCS becomes ineffective, leading to degraded statistics for these systems.

Figure 7 shows the bias of DCS occurrences according to their lifetime duration between the five degraded runs of TOOCAN and the TOOCAN reference run. Up to 2 h of consecutive missing images, results indicate a bias which tends toward 0 % for the longest ones. For a run performed with 3 h of consecutive missing images, we observe an over-estimation of the DCS occurrences around 20 % for systems longer than 10 h. The biases on DCS occurrences increase drastically with the 4 h and 5 h consecutive missing images, and we observe an over-estimation of DCS occurrences greater than 80 % and 100 % respectively for long DCS lifetime durations. A maximum of 3 h of consecutive missing images therefore seems to be a good compromise for ensuring tracking continuity and minimizing the impact on DCS morphological parameters of longest-lived systems. Note that here by perturbing every day the dataset, we are exploring the worst-case scenario similar to the MFG eclipse' seasons.

**3.3 TOOCAN Implementation**

The TOOCAN algorithm has been applied on the harmonized infrared observations of each geostationary platform at a 30min temporal resolution from January 2012 to December 2020. The spatial coverage of each monitored region described table 2 is wide enough in longitude to offer an overlapping area with its neighbor's regions and is extended from 40° S to 40° N in latitude to avoid an impact of the image boundaries on the tracking of DCS over the tropical belt. Applying a minimum buffer strip of 5° from the minimum/maximum geographical coordinates of these extended regions, ensure that the shape and trajectories of convective systems identified in the tropical belt are not impacted by the image boundaries.

The time series of the number of days per month impacted by 1 h, 2 h, 3 h and more than 3 h successive missing images over the 2012-2020 period is presented Fig. 8 for each geostationary platform and region of interest. During this period, 1.5 % of days are impacted by consecutive missing data lasting more than 3h over the entire tropics and for all the geostationary platforms. From 2012 to 2017, METEOSAT-7 is the most strongly impacted by consecutive missing data especially from early August to mid-September and from February to March which is explained by solar eclipses. During these periods, more than 5.8 % of the days are impacted by a maximum of 3 h consecutive missing images and 0.73 % by a sequence of more than 3 h consecutive missing images. From the analysis carried out in section 3.2.b, we define a maximum of 3h consecutive missing images, above which the tracking of DCS will be stopped. This duration of missing images is a good trade-off to track convective systems with a reasonable statistical bias on their occurrences as seen previously but also with a limited number of days impacted by such successive missing data events.

Finally, in order to have a homogeneous analysis of DCS thereafter, the tracking process is not carried out on the southern scan of the MTSAT-1 and 2 platform from January 2012 to May 2015, as the 30-minute time-frequency requirement is not fulfilled. For similar reason, TOOCAN is not applied on a little region between 118° W and 108° W in the southern hemisphere which is monitored by GOES-15 only every 3 h.

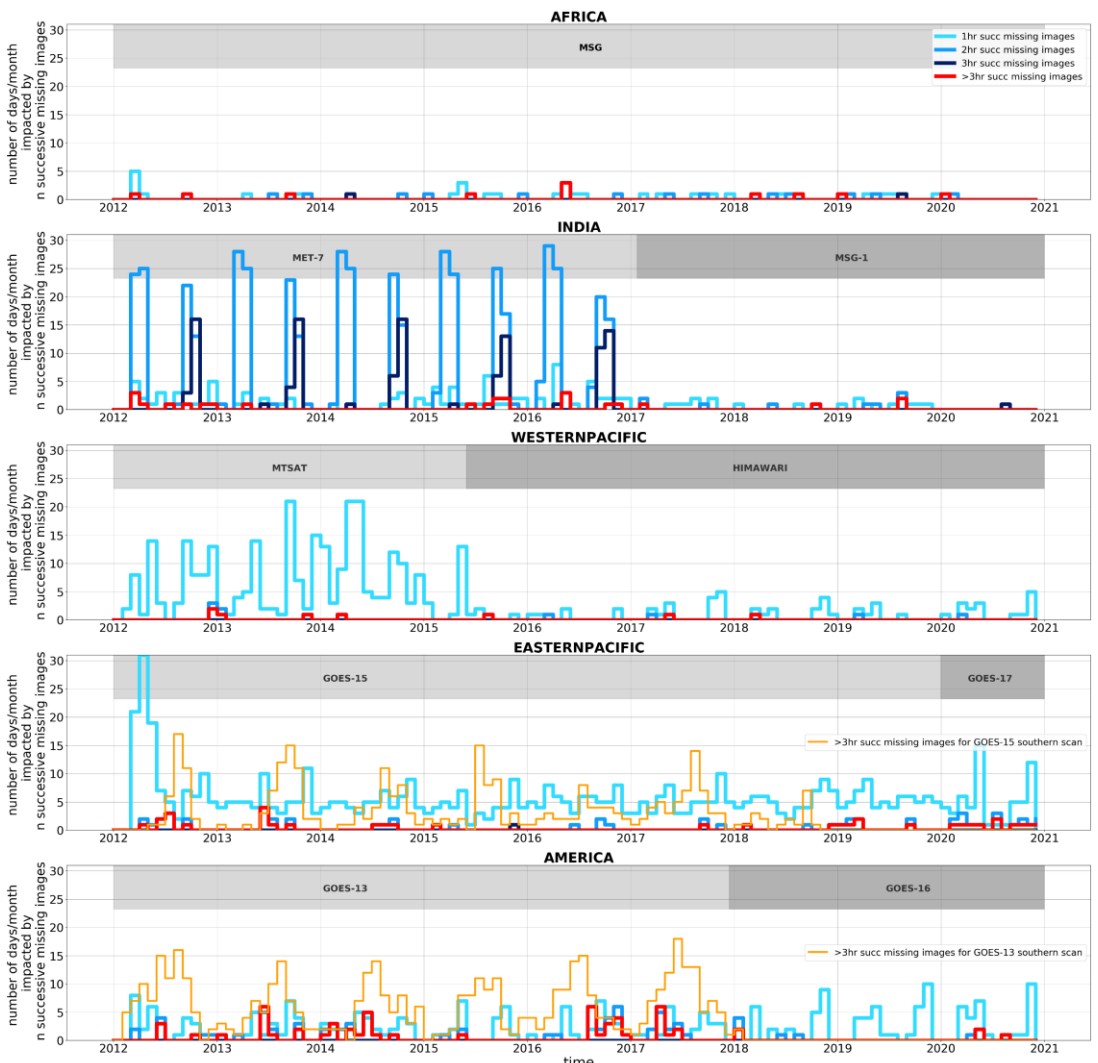

**Figure 8** Time series of the number of days per month impacted by 1h, 2h, 3h and more than 3h successive missing images between 2012 and 2020 and for the 5 regions of interest. The grey shading indicates the operability period for each geostationary platform.

**4 The TOOCAN and the CACATOES datasets**

**4.2 TOOCAN dataset**

The processing has given rise to a 9-year of DCS database documenting convective systems identified over the tropical belt. From one side, the dataset documents the integrated morphological parameters of each identified DCS (Table B1). On the other side, the morphological properties of each convective system are described every 30 minutes along their life cycles (Table B2). Systems impacted by region boundaries, as well as systems interrupted by consecutive images events, or initiated
from fresh start are flagged. A quality flag lower than 11110 is a good compromise to keep DCS little impacted by imagery issues. The first digit of the quality flag indicates whether a DCS is born naturally or due to consecutive missing images, while the second digit indicates whether a DCS dissipates naturally or as a result of consecutive missing images. The third digit reveals if a DCS is impacted by image boundaries or missing pixels, with a value greater than one indicating such an issue. Values of these three digits greater than one indicate problems in initiation, dissipation, or touching the edge of the image. The
last two digits of the quality flag represent the number of missing images during the lifecycle of a given DCS.

Each cloud system has received a unique label and the documentation of the convective systems has also been enhanced with some classifications. First, convective systems have been classified following the three categories of systems introduced in Fiolleau and Roca (2013a) and Roca et al. (2017a). The Deep convective systems are also categorized according to their organization, their shapes at small and large scales. In an identical way to the categories introduced by Maddox (1980) and
Jirak et al. (2003), DCS are then classified into four types as Mesoscale Convective Complex (MCC), Persistent Elongated Convective System (PECS), Meso-β-Circular Convective System (MβCCS), and Meso-β-Elongated Convective System (MβECS) (table 3). Finally, a last classification is performed by associating the DCS with the synoptic tropical storm recorded in the IBTrACS database. As introduced in (Hennon et al., 2011), a system located within a 1000 km radius of a cyclone is flagged according the storm type and the SSHS category (table 3).

The Analyses of DCS database can be conducted regions per regions. However, to focus on the whole tropical belt, we restrict the spatial coverage for each region to prevent double counting a same DCS identified in two adjacent geostationary platforms. Table 4 defines the spatial coverages to apply on each region of interest according to the configuration of the GEOring, which evolves along the period. A quality control indicator is associated to each identified DCS to indicate whether the cloud system has been impacted by interruptions/restarts, image edges, or missing images. Hence, 0.35 % of the total DCS population is
found to be impacted by recovery and interruption of the tracking algorithm, with a maximum for the American (1.76 %) and Eastern Pacific regions (1.47 %), explained by the rapid scan operation modes of GOES-13 and GOES-15.

By filtering DCS which do not pass this quality control, a total of around $15 \times 10^6$ DCS has been identified and tracked by TOOCAN from 2012 to 2020 over the entire tropical belt (30° S-30° N), $\sim 9 \times 10^6$ over the oceans, $\sim 4.1 \times 10^6$ over the continents and $1.7 \times 10^6$ over coastal regions. Oceanic convective systems are described by a slightly longer average lifetime duration
(6.25 h) than continental systems (6 h). Oceanic systems can last up to 102h, while the continental ones reach a maximum of

43.5 h. A large majority of convective systems are characterized by a maximum area between $1 \times 10^3$ and $2 \times 10^5$ km$^2$, but some of them can reach a maximum extent up to $2.3 \times 10^6$ km$^2$ over the ocean and $1.3 \times 10^6$ km$^2$ over continents.

| DCS Classifications | Definition | Contribution to population (%) | Contribution to cold cloudiness (%) | references |
|---|---|---|---|---|
| Class 1 | - DCS with a lifetime duration < 5h | 36.7 | 5.5 | Roca et al. (2017) |
| Class 2a | DCS with a lifetime duration > 5h and describing only one maximum of their surface along their life cycle | 52.8 | 84 | Roca et al. (2017) |
| Class2b | DCS with a lifetime duration > 5h and describing several maximum of their surface along their life cycle | 10.5 | 10.5 | Roca et al. (2017) |
| MCC | - Cold cloud region ≤-52° with area ≥ 50 000 km²<br>- Size definition met for ≥ 6 h<br>- Eccentricity > 0.7 at time of maximum extent | 0.08 | 3.3 | Jirak et al. (2003) |
| PECS | - Cold cloud region ≤-52° with area ≥ 50 000 km²<br>- Size definition met for ≥ 6 h<br>- 0.2 ≤ Eccentricity < 0.7 at time of maximum extent | 0.08 | 3 | Jirak et al. (2003) |
| MβCCS | - Cold cloud region ≤-52° with area ≥ 50 000 km²<br>- Size definition met for ≥ 6 h<br>- Eccentricity > 0.7 at time of maximum extent | 0.3 | 3.9 | Jirak et al. (2003) |
| MβECS | - Cold cloud region ≤-52° with area ≥ 50 000 km²<br>- Size definition met for ≥ 6 h<br>- 0.2 ≤ Eccentricity < 0.7 at time of maximum extent | 0.41 | 5.36 | Jirak et al. (2003) |
| MCC | - Cold cloud shield ≤-32° with area ≥ 100 000 km²<br>- Interior Cold cloud region ≤-52° with area ≥ 50 000 km²<br>- Size definition met for ≥ 6 h<br>- Eccentricity > 0.7 at time of maximum extent | 0.04 | 2.31 | Maddox (1980) |
| Mixture | | 0.03 | 0.03 | |
| Not reported | | 0.77 | 1.2 | |
| subtropical | | 0.27 | 0.34 | |
| Extratropical | | 0.01 | 0.01 | |
| Disturbance | | 0.01 | 0.01 | Knapp et al. (2011) |
| Tropical | - DCS located at a maximum distance of 1000km from a tropical storm identified and saved into the IBTrACS database | 3 | 5 | Knapp et al. (2011) |
| SSHS category 1 | | 0.41 | 0.82 | Knapp et al. (2011) |
| SSHS category 2 | | 0.2 | 0.41 | Knapp et al. (2011) |
| SSHS category 3 | | 0.22 | 0.46 | Knapp et al. (2011) |
| SSHS category 4 | | 0.2 | 0.46 | Knapp et al. (2011) |
| SSHS category 5 | | 0.01 | 0.04 | Knapp et al. (2011) |

**Table 3** DCS classifications and physical characteristics used in the TOOCAN database

Regarding the system classifications, systems belonging to Class 1 contribute to 36.7 % of the total population but only 5.5 % of the cold cloudiness area (table 3). Class 2a convective systems contribute to 52.8 % of the total population and 84% of the cold cloudiness area, consistent with the results obtained in Roca et al. (2017a). Convective systems classified as MCC according to the definition given by Jirak et al. (2003), represent only 0.08 % of the total population but contribute to 3.3 % of the cold cloudiness area. Finally, a total of 156594 DCS are included within a 1000 km radius of a tropical cyclone (1 % of

the total population), and only 2081 of them are associated to a category 5 cyclone (SSHS).

| | Region | Period | | | | | |
|---|---|---|---|---|---|---|---|
| | | 01/2012 05/2015 | 06/2015 12/2016 | 01/2017 12/2017 | 01/2018 11/2018 | 12/2018 12/2019 | 01/2020 12/2020 |
| Effective spatial coverage | AFRICA | 37,5°W-37,5°E 30°S-30°N | 37,5°W-37,5°E 30°S-30°N | 37,5°W-37,5°E 30°S-30°N | 37,5°W-37,5°E 30°S-30°N | 37,5°W-37,5°E 30°S-30°N | 37,5°W-37,5°E 30°S-30°N |
| | INDIA | 37,5°E-101,5°E 30°S-30°N | 37,5°E-101,5°E 30°S-30°N | 37,5°E-91,5°E 30°S-30°N | 37,5°E-91,5°E 30°S-30°N | 37,5°E-91,5°E 30°S-30°N | 37,5°E-91,5°E 30°S-30°N |
| | WESTERNPACIFIC | 101,5°E-184,5°E 30°S-30°N | 101,5°E-184,5°E 30°S-30°N | 91,5°E-184,5°E 30°S-30°N | 91,5°E-184,5°E 30°S-30°N | 91,5°E-184,5°E 30°S-30°N | 91,5°E-184,5°E 30°S-30°N |
| | EASTERNPACIFIC | 184,5°E-251,5°E 30°S-30°N | 184,5°E-251,5°E 30°S-30°N | 184,5°E-251,5°E 30°S-30°N | 184,5°E-240,5°E 30°S-30°N | 184,5°E-240,5°E 30°S-30°N | 184,5°E-240,5°E 30°S-30°N |
| | AMERICA | 108.5°W-37.5°W 30°S-30°N | 108.5°W-37.5°W 30°S-30°N | 108.5°W-37.5°W 30°S-30°N | 119.5°W-37.5°W 30°S-30°N | 119.5°W-37.5°W 30°S-30°N | 119.5°W-37.5°W 30°S-30°N |

**Table 4** Effective spatial coverage to be applied to each region and for the periods corresponding to specific GEOring configurations for a tropical belt analysis

### 4.2.1 An illustration of the TOOCAN database

The annual climatology of the occurrence and morphological characteristics of deep convective systems is shown Fig. 9 in a phase diagram using lifetime duration and maximum extent as coordinates. Here, DCS within a 1000 km radius of a tropical cyclone have been removed from this analysis.

The distribution is described by systems lasting few hours and reaching around 1000km² to systems lasting several days and reaching up to few millions km² (Figure 9a). While a strong relationship is observed between lifetime duration and the

465 maximum extent at first order, it is to also be noticed that a same lifetime duration can be associated to a large spread of maximum extent. Fig. 9b shows also that the larger the cloud shield, the colder the temperatures at the top of the cloud and therefore the deeper the cloud. Similar to Roca et al. (2024), both land and ocean distributions are described by a "V" pattern with two branches associated with the warmer systems. The shape of the cloud shield of the system is shown in Fig. 9c, with the distribution of the eccentricity of the cloud shield at the time of maximum extent along the life cycle. The eccentricity is

470 defined by ratio of semi-minor axis to semi-major axis of the equivalent ellipse. While the largest and coldest deep convective systems are characterized by circularity of their cold cloud shields (eccentricity > 0.7), the warmer DCS located in the two branches of the "V" pattern are more characterized by linear shapes of their cold cloud shields (eccentricity < 0.5). The

distribution of the average speed according to maximum extent and lifetime duration is shown Fig. 9d. The fastest systems are found in the upper part of the distribution, while the slowest are found in its lower part. These features are more pronounced over the ocean compared to land.

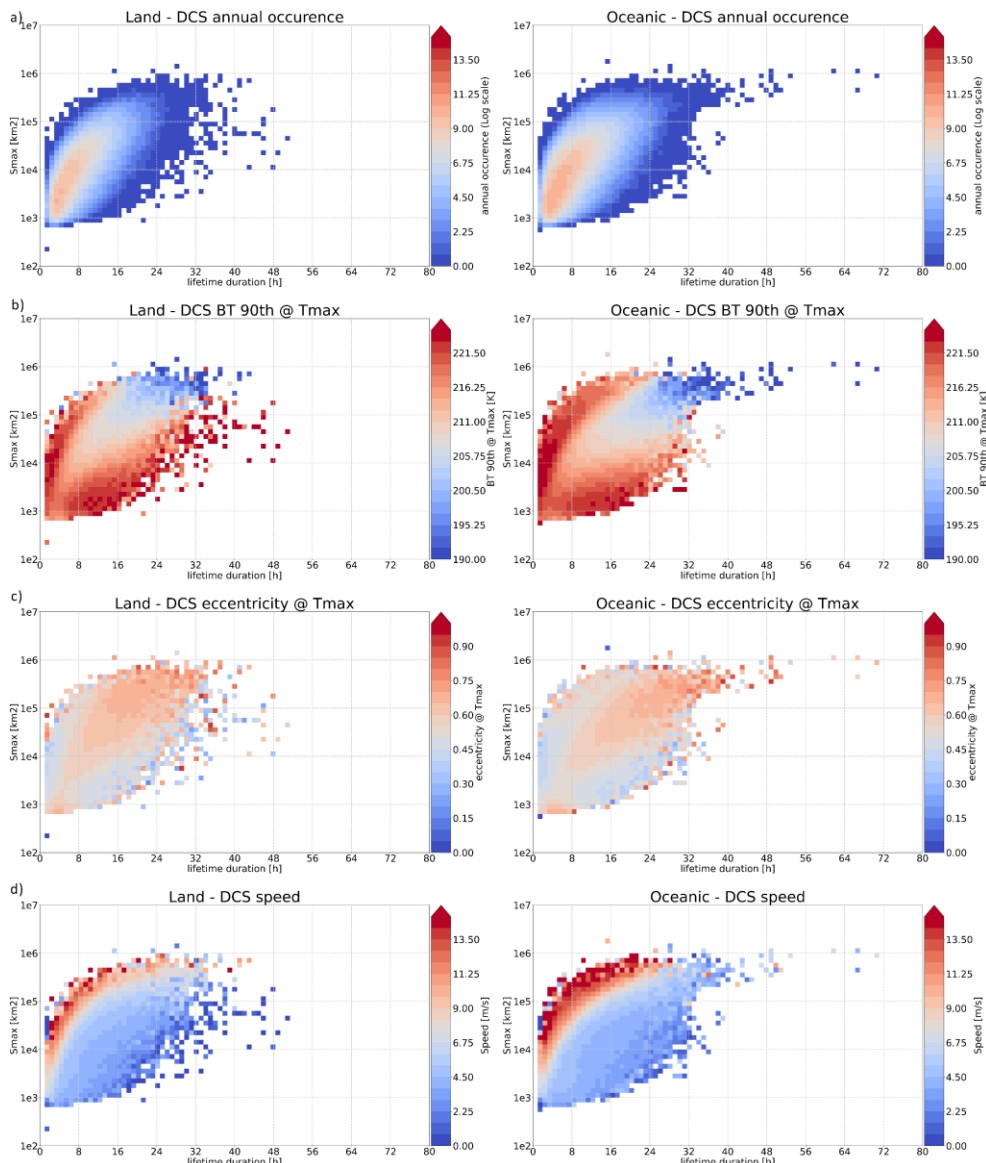

**Figure 9** (a) annual occurrence of the deep convective systems for continental (left) and the oceanic region; (b) 90[th] percentile of the cluster brightness temperature at the time of maximum extent; (c) eccentricity of the cluster at the time of maximum extent; (d) movement speed of the deep convective system

### 4.3 CACATOES dataset

The CACATOES database is a level-3 product derived from the TOOCAN database allowing a Eulerian view of the deep convective systems properties from a grid box perspective. The method was introduced and used in several studies (Roca and Fiolleau 2020, Berthet et al., 2017, Roca et al., 2014) and makes it easier the joint analysis with auxiliary data gridded on the same daily 1° × 1° grid box. The integrated morphological parameters of each DCS is gridded into a 1° × 1° daily grid (Table C1). For that, the full resolution pixels composing the convective systems identified within the TOOCAN segmented images are projected into each daily 1° × 1° grid box. The cold cloudiness fraction of each DCS which overpasses a given daily grid box is computed, and their morphological properties are assigned to that particular grid box. If cold cloud shield of a given DCS overpasses more than one daily grid box, the cold cloudiness of this given DCS is hence distributed onto each of the associated daily grid boxes. Similarly, when a system is lasting more than a day, the associated cold cloudiness is distributed over all the relevant days. Owing to the DCS propagation, cold cloud surfaces and lifetime durations, several systems can overpass a same 1° × 1° grid box during one day. In that case, they contribute all together to the total cold cloudiness of this given grid box. It has been defined that a maximum of 25 individual systems can overpass each grid box in a day. Within a given daily 1°x1° grid box, the DCS morphological properties are finally sorted according to their cold cloudiness occupation, so that the most representative DCS can be easily identified. Roca and Fiolleau (2020) have shown that a couple of DCS impacts significantly each daily 1°x1° grid box, while most of the other systems have very small contributions to the cold cloudiness. The statistical analysis of the DCS morphological parameters requires special cares when considered in a Eulerian framework. For instance, a DCS with a long lifetime duration may overpass only a few moments and a small footprint of a daily 1°x1° grid box skewing the statistical results. Therefore, the average of a given morphological parameter over a region and time period has to be weighted using the actual cloud occupation of each system within each grid box. Similarly, the computation of the DCS population over a region and time period has to be considered with caution and can be computed by the sum of all the systems cold cloud fractions overpassing the daily grid boxes.

### 4.3.1 An illustration of the CACATOES database

Figure 10 shows the daily 1°x1° spatial distribution of the DCS density calculated from January 2012 to December 2020 over the entire tropical belt. The hatched areas indicate that the southern part of the Western and a southern band between 118° W and 108° W of the Eastern Pacific regions have been affected by missing data respectively from January 2012 to May 2015 and January 2012 to November 2017 respectively. The DCS density of these two specific regions have then been computed with a lower number of days involved. This may impact the patterns of DCS density, and therefore we would like to emphasize to future users that the analysis should be carried out with caution. Also, note that the map was built considering only days not impacted by any tracking interruptions over the any part of the tropical region and corresponding to 61.5 % of the total number of days over the entire 2012-2020 period. This number is mainly explained by the interruption of the southern scans of GOES-13 and GOES-15 due to Rapid Scan Operations

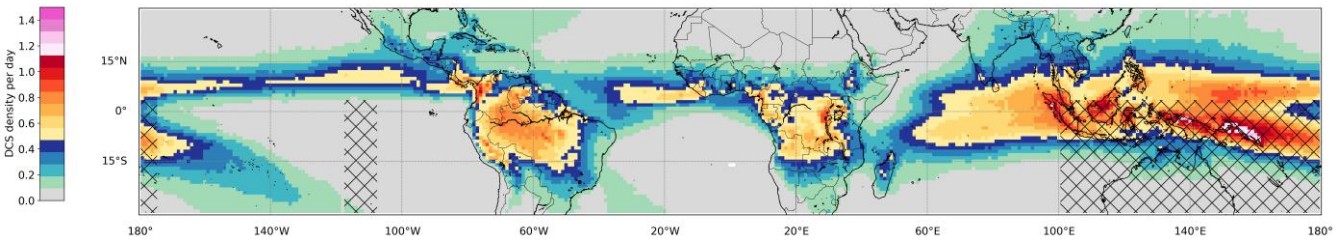

**Figure 10** Map of the deep convective systems density per day for a 1°x1° grid box from January 2012 to December 2020. The hatched areas indicate missing data over the southern part of the Western and Eastern Pacific regions from January 2012 to May 2015 and from January 2012 to November 2017 respectively.

The geographical distribution of DCS density per day is consistent with previous DCS climatology based on other definitions, algorithms and satellite observations, either from local or global studies (Mohr and Zipser 1996, Liu et al. 2008, Feng et al. 2021, Huang et al. 2018). A zonal and homogeneous structure occurs over the Atlantic Ocean extending from the Guinean coast to 40°W (Machado et al., 1992). The Indian Ocean shows a large and zonally structured region, corresponding to the ITCZ (Roca and Ramanathan, 2000). On the continent, the West African deep convective systems extend no further north than 17°N, and the Indian Ocean convective systems extend to the foot of the Himalayas. DCS are also numerous over the western half of the maritime continent (Williams and Houze, 1987) with similar occurrence to that of Southeast Asia and the Philippines. The eastern Pacific ITCZ is also characterized by a large occurrence of DCS.

## 5 Data Format

### 5.1 TOOCAN Data format

The TOOCAN database is composed by two types of files. Regional TOOCAN segmented images at a 0.04° spatial resolution are produced every 30-minutes in a NetCDF-4 format with metadata following the Climate and Forecast (CF) Convention version 1.6 and Attribute Convention for Dataset Discovery (ACDD) version 1.3. The TOOCAN segmented images files contain the following information:

- **DCS_number**: labelled pixels of the convective systems identified by the TOOCAN algorithm.
- **latitude**: the latitude values of the grid, in degrees, ranging between (-40° N,40° N)
- **longitude**: the longitude values of the grid, in degrees
- **time**: the starting time scan of the image in seconds, since 1st January 1970
- **scantime**: time in seconds since 1st January 1970 at which each line of the TOOCAN segmented image is scanned by the geostatellite platform.

Regional and monthly tracking files are produced in a NETCDF-4 format with metadata following the Climate and Forecast (CF) Convention version 1.6 and Attribute Convention for Dataset Discovery (ACDD) version 1.3 to document the DCS integrated morphological parameters as well as the DCS parameters at each 30minute-step of their life cycles. Note that similar regional and monthly tracking files have also been produced in an ASCII format, in order to ensure a continuity with previous versions of TOOCAN databases

Only the deep convective systems which initiate in a given month are stored in the corresponding monthly tracking file. If a DCS is initiated in a given month, but its dissipation extends beyond the end of that same month, the complete life cycle of this DCS is recorded in the file for the month corresponding to its birth. Each DCS is described by a unique label, and the link can be easily established between a given DCS described in a monthly tracking file and the pixels constituting this given DCS within the TOOCAN segmented images.

## 5.2 CACATOES Data Format

The 1°x1°-1day CACATOES tropical monthly files, describing the characteristics of DCS overpassing the daily 1° x1° lon/lat grid boxes, are also produced in a NetCDF-4 format with metadata following the Climate and Forecast (CF) Convention version 1.6 and Attribute Convention for Dataset Discovery (ACDD) version 1.3

## 5.3 Data availability

The TOOCAN database is available over the 2012-2020 period for each region of interest with a 40° S-40° N latitudinal coverage (Eastern-Pacific, America, Africa, India, Western-Pacific) in the following link: https://doi.org/10.14768/1be7fd53-8b81-416e-90d5-002b36b30cf8 (Fiolleau and Roca, 2023). The CACATOES database, derived from the TOOCAN dataset, is available for the 2012-2020 period over the whole tropical belt [30°S-30°N] in the following link: https://doi.org/10.14768/98569eea-d056-412d-9f52-73ea07b9cdca (Fiolleau and Roca, 2023). The 2012-2020 homogenized infrared geostationary level-1C dataset described in this paper, and on which has been applied the TOOCAN algorithm can be accessed via the repository under the following data DOI: https://doi.org/10.14768/93f138f5-a553-4691-96ed-952fd32d2fc3 (Fiolleau and Roca, 2023). The DOI landing pages provide the up-to-date information on how to access the database as well as a number of useful references for users.

## 6 Conclusions and outlook

A unique database of the deep convective systems and their morphological characteristics covering the 2012-2020 period over the intertropical belt has been introduced. The DCS morphology is obtained thanks to the TOOCAN tracking algorithm applied on a homogenized GEOring infrared archive. The homogenized GEOring database has been built from level-1 data of a fleet

of geostationary platforms originating from various sources. The temporal and spatial resolution of this GEOring archive is respectively 30 minutes and 0.04°. The GEOring dataset has been further inter-calibrated, spectrally adjusted, and limb darkening corrected, specifically for the high cold cloud shield, onto a common reference, the IR channel of the ScaRaB radiometer on-board Megha-Tropiques following the methodology introduced in Fiolleau et al. (2020). The global

homogeneity of the IR GEOring dataset is then characterized by residual error of 1.33 K. Over the 9-year period, the configuration of the geostationary fleet nevertheless drastically changes. In June 2015, MTSAT is then replaced by HIMAWARI-8 over the Western Pacific Ocean. In January 2017, the end of operation of METEOSAT-7 corresponds to the arrival of MSG-1. Finally GOES-16 and GOES-17 have become operational respectively in December 2017 over the American and in January 2020 for the East Pacific Ocean.

An assessment of the sensitivity of the DCS identified by TOOCAN to the radiometric errors of the homogenized GEOring has been carried out. This analysis has shown a very small impact of a 1.5 K residual error on the DCS occurrences whatever their lifetime durations. Similarly, we have evaluated the impact of consecutive missing images on the quality of the DCS tracking. By filling the missing data periods by the available IR data just before and after the missing data gap, we have shown that up to 3 h period of consecutive missing images, the impact is relatively small on the DCS occurrences. Hence, the

occurrence of systems lasting more than 10h is skewed by 20 % for a 3 h period of consecutive missing images. Beyond 3 h of consecutive missing images, the impact on the DCS segmentation is too high, and the tracking process has to be stopped.

The TOOCAN algorithm has then been processed on the homogenized GEOring IR data over the 2012-2020 period and on the latitude band 40° S-40° N. The resulting database gives an access to the integrated morphological parameters of each DCS (Location and time of initiation and dissipation, lifetime duration, propagated distance, cold cloud maximum extent…), as well

as the evolution of the morphological properties along the DCS life cycles. The DCS located near a cyclone identified in the IBTrACS database (Knapp et al., 2010) have been flagged. A total of $15 \times 10^6$ DCS have been detected and tracked by TOOCAN over the tropical regions and the 9-year period. The analysis of DCS database over the tropical oceans and continents shows the large variety of DCS characteristics and organization encountered. DCS can last few hours up to several days, and are distributed by cloud surfaces from 1000 km² to few millions of km². Oceanic DCS are described by a longer lifetime duration

and larger cold cloud surfaces. Over the both regions, while a strong relationship is observed between lifetime duration and Maximum surface extent in a first order, we can also notice a large spectrum of maximum extent for a given lifetime duration. The 2D spatial distribution of DCS density over the Tropics is also in line with previous DCS climatology produced from other formulation of tracking algorithms and geostationary IR dataset (Feng et al. 2021, Huang et al. 2018, Rajagopal et al. 2023).

## Appendix A

An example of a header of the netCDF-4 file for the TOOCAN monthly tracking file.

dimensions:

    DCS = 40915 ;

time = UNLIMITED ; // (1523 currently)

variables:

    int time(time) ;

        time:units = "seconds since 1970-01-01" ;

        time:long_name = "time" ;

int DCS(DCS) ;

        DCS:units = "none" ;

        DCS:long_name = "Label of the Deep Convective Systems" ;

    int INT_DCSnumber(DCS) ;

        INT_DCSnumber:_FillValue = -999 ;

INT_DCSnumber:units = "" ;

        INT_DCSnumber:long_name = "Label of the DCS in the TOOCAN segmented images" ;

    int INT_DCS_qualitycontrol(DCS) ;

        INT_DCS_qualitycontrol:_FillValue = -999 ;

        INT_DCS_qualitycontrol:units = "" ;

INT_DCS_qualitycontrol:long_name = "Quality control on the DCS initiation/dissipation..." ;

    int INT_classif(DCS) ;

        INT_classif:_FillValue = -999 ;

        INT_classif:units = "" ;

        INT_classif:long_name = "Classification of the DCS according to Roca etal (2017)" ;

INT_classif:flag_values = 1, 2, 3 ;

        INT_classif:flag_meanings = "DCS with a duration < 5hr, DCS with a duration ≥ 5hr and described by a single maximum of their cold surfaces along their life cycles, DCS with a duration ≥ 5hr and described by several maximums of their cold surfaces along their life cycles" ;

```
        float INT_duration(DCS) ;

                INT_duration:_FillValue = -999.f ;

INT_duration:units = "hr" ;

                INT_duration:long_name = "DCS lifetime duration" ;

        int INT_UTC_timeInit(DCS) ;

                INT_UTC_timeInit:_FillValue = -999 ;

                INT_UTC_timeInit:units = "seconds since 1st January 1970" ;

INT_UTC_timeInit:long_name = "Universal Time of the DCS initiation " ;

        int INT_localtime_Init(DCS) ;

                INT_localtime_Init:_FillValue = -999 ;

                INT_localtime_Init:units = "seconds since 1st January 1970" ;

                INT_localtime_Init:long_name = "Local time of the DCS initiation initiation" ;

float INT_lonInit(DCS) ;

                INT_lonInit:_FillValue = -999.f ;

                INT_lonInit:units = "degrees" ;

                INT_lonInit:long_name = "Longitude of the DCS center of mass at its initiation" ;

        float INT_latInit(DCS) ;

INT_latInit:_FillValue = -999.f ;

                INT_latInit:units = "degrees" ;

                INT_latInit:long_name = "Latitude of the DCS center of mass at its initiation" ;

        int INT_UTC_timeEnd(DCS) ;

                INT_UTC_timeEnd:_FillValue = -999 ;

INT_UTC_timeEnd:units = "seconds since 1st January 1970" ;

                INT_UTC_timeEnd:long_name = "Coordinated Universal Time of the DCS dissipation" ;

        int INT_localtime_End(DCS) ;

                INT_localtime_End:_FillValue = -999 ;

                INT_localtime_End:units = "seconds since 1st January 1970" ;
```

INT_localtime_End:long_name = "Local time of the DCS dissipation" ;

     float INT_lonEnd(DCS) ;

        INT_lonEnd:_FillValue = -999.f ;

        INT_lonEnd:units = "degrees" ;

        INT_lonEnd:long_name = "Longitude of the DCS center of mass at its dissipation" ;

float INT_latEnd(DCS) ;

        INT_latEnd:_FillValue = -999.f ;

        INT_latEnd:units = "degrees" ;

        INT_latEnd:long_name = "Latitude of the DCS center of mass at its dissipation" ;

     float INT_velocityAvg(DCS) ;

INT_velocityAvg:_FillValue = -999.f ;

        INT_velocityAvg:units = "m/s" ;

        INT_velocityAvg:long_name = "Average velocity of the DCS from its initiation to its dissipation" ;

     float INT_distance(DCS) ;

        INT_distance:_FillValue = -999.f ;

INT_distance:units = "km" ;

        INT_distance:long_name = "DCS progagated distance" ;

     float INT_lonmin(DCS) ;

        INT_lonmin:_FillValue = -999.f ;

        INT_lonmin:units = "degrees" ;

INT_lonmin:long_name = "Minimum longitude of the DCS along its life cycle" ;

     float INT_lonmax(DCS) ;

        INT_lonmax:_FillValue = -999.f ;

        INT_lonmax:units = "degrees" ;

        INT_lonmax:long_name = "Maximum latitude of the DCS along its life cycle" ;

float INT_latmin(DCS) ;

        INT_latmin:_FillValue = -999.f ;

INT_latmin:units = "degrees" ;

INT_latmin:long_name = "Minimum longitude of the DCS along its life cycle" ;

float INT_latmax(DCS) ;

INT_latmax:_FillValue = -999.f ;

INT_latmax:units = "degrees" ;

INT_latmax:long_name = "Maximum latitude of the DCS along its life cycle" ;

float INT_tbmin(DCS) ;

INT_tbmin:_FillValue = -999.f ;

INT_tbmin:units = "K" ;

INT_tbmin:long_name = "Minimum brightness temperature of the DCS along its life cycle" ;

int INT_surfmaxPix_235K(DCS) ;

INT_surfmaxPix_235K:_FillValue = -999 ;

INT_surfmaxPix_235K:units = "number of pixels" ;

INT_surfmaxPix_235K:long_name = "Maximum cold cloud surface at 235K reached by the DCS along its life cycle" ;

float INT_surfmaxkm2_235K(DCS) ;

INT_surfmaxkm2_235K:_FillValue = -999.f ;

INT_surfmaxkm2_235K:units = "km2" ;

INT_surfmaxkm2_235K:long_name = "Maximum cold cloud surface at 235K reached by the DCS along its life cycle " ;

float INT_surfmaxkm2_220K(DCS) ;

INT_surfmaxkm2_220K:_FillValue = -999.f ;

INT_surfmaxkm2_220K:units = "km2km2" ;

INT_surfmaxkm2_220K:long_name = "Maximum cold cloud surface at 220K reached  by the DCS along its life cycle" ;

float INT_surfmaxkm2_210K(DCS) ;

INT_surfmaxkm2_210K:_FillValue = -999.f ;

INT_surfmaxkm2_210K:units = "km2" ;

INT_surfmaxkm2_210K:long_name = "Maximum cold cloud surface at 210K reached  by the DCS along its life cycle" ;

float INT_surfmaxkm2_200K(DCS) ;

```
        INT_surfmaxkm2_200K:_FillValue = -999.f ;

INT_surfmaxkm2_200K:units = "km2" ;

        INT_surfmaxkm2_200K:long_name = "Maximum cold cloud surface at 200K reached  by the DCS along its life cycle" ;

    float INT_surfcumkm2_235K(DCS) ;

        INT_surfcumkm2_235K:_FillValue = -999.f ;

        INT_surfcumkm2_235K:units = "km2" ;

INT_surfcumkm2_235K:long_name = "Cumulated cold cloud surface at 235K along the DCS life cycle" ;

    int INT_classif_JIRAK(DCS) ;

        INT_classif_JIRAK:_FillValue = -999 ;

        INT_classif_JIRAK:units = "none" ;

        INT_classif_JIRAK:long_name = "DCS classification according to the JIRAK definition (Jirak etal (2003)" ;

INT_classif_JIRAK:flag_values = 0, 1, 2, 3, 4 ;

        INT_classif_JIRAK:flag_meanings = "no classification,MCC,PECS,MBCC,MBECC" ;

    int INT_classif_MADDOX(DCS) ;

        INT_classif_MADDOX:_FillValue = -999 ;

        INT_classif_MADDOX:units = "none" ;

INT_classif_MADDOX:long_name = "DCS classification according to the MADDOX definition Maddox (1980)" ;

        INT_classif_MADDOX:flag_values = 0, 1, 2, 3, 4, 5, 6, 7, 8, 9, 10, 11, 12, 13, 14, 15 ;

        INT_classif_MADDOX:flag_meanings = "no matching with TS, Mixture, Not reported, disturbance, subtropical storm, extratropical storm, tropical
    storm, cyclone SSHS category 1, cyclone SSHS category 2, cyclone SSHS category 3, cyclone SSHS category 4, cyclone SSHS category 5" ;

    int INT_TS_number_IBTRACS(DCS) ;

INT_TS_number_IBTRACS:_FillValue = -999 ;

        INT_TS_number_IBTRACS:units = "none" ;

        INT_TS_number_IBTRACS:long_name = "number of the Tropical Storm in the IBTRACS database associated with the DCS within a 1000km
    radius" ;

    int INT_TS_nature_IBTRACS(DCS) ;

INT_TS_nature_IBTRACS:_FillValue = -999 ;

        INT_TS_nature_IBTRACS:units = "none" ;
```

INT_TS_nature_IBTRACS:long_name = "nature of the Tropical Storm in the IBTRACS database" ;

float INT_TS_mindistance_IBTRACS(DCS) ;

INT_TS_mindistance_IBTRACS:_FillValue = -999.f ;

735   INT_TS_mindistance_IBTRACS:units = "km2" ;

INT_TS_mindistance_IBTRACS:long_name = "Distance of the DCS to the Tropical Storm (maximum distnace: 1000km)" ;

int QCgeo_IRimage(time) ;

QCgeo_IRimage:_FillValue = -999 ;

QCgeo_IRimage:units = "nodimension" ;

740   QCgeo_IRimage:long_name = "Quality control on the GEO IR data" ;

QCgeo_IRimage:flag_values = 0, 1, 2 ;

QCgeo_IRimage:flag_meanings = "Missing GEO IR data, The Full GEO IR data OK, The Only North scan of GEO IR data OK" ;

float LC_tbmin(DCS, time) ;

LC_tbmin:_FillValue = -999.f ;

745   LC_tbmin:units = "K" ;

LC_tbmin:long_name = "Minimum brightness temperature" ;

float LC_tbavg_235K(DCS, time) ;

LC_tbavg_235K:_FillValue = -999.f ;

LC_tbavg_235K:units = "K" ;

750   LC_tbavg_235K:long_name = "Average brightness temperature at 235K" ;

float LC_tbavg_208K(DCS, time) ;

LC_tbavg_208K:_FillValue = -999.f ;

LC_tbavg_208K:units = "K" ;

LC_tbavg_208K:long_name = "Average brightness temperature at 208K" ;

755  float LC_tbavg_200K(DCS, time) ;

LC_tbavg_200K:_FillValue = -999.f ;

LC_tbavg_200K:units = "K" ;

LC_tbavg_200K:long_name = "Average brightness temperature at 200K" ;

```
float LC_tb90th(DCS, time) ;
LC_tb90th:_FillValue = -999.f ;

            LC_tb90th:units = "K" ;

            LC_tb90th:long_name = "Tb 90th percentile" ;

        int LC_UTC_time(DCS, time) ;

            LC_UTC_time:_FillValue = -999 ;

LC_UTC_time:units = "seconds since 1st January 1970" ;

            LC_UTC_time:long_name = "Coordinated Universal Time of the DCS " ;

        int LC_localtime(DCS, time) ;

            LC_localtime:_FillValue = -999 ;

            LC_localtime:units = "seconds since 1st January 1970" ;

LC_localtime:long_name = "Local Time of the DCS " ;

        float LC_lon(DCS, time) ;

            LC_lon:_FillValue = -999.f ;

            LC_lon:units = "degrees" ;

            LC_lon:long_name = "longitude of the DCS center of mass" ;

float LC_lat(DCS, time) ;

            LC_lat:_FillValue = -999.f ;

            LC_lat:units = "degrees" ;

            LC_lat:long_name = "latitude of the DCS center of mass" ;

        int LC_x(DCS, time) ;

LC_x:_FillValue = -999 ;

            LC_x:units = "pixels" ;

            LC_x:long_name = "Column of the DCS center of mass" ;

        int LC_y(DCS, time) ;

            LC_y:_FillValue = -999 ;

LC_y:standard_name = "pixels" ;
```

LC_y:long_name = "Line of the DCS center of mass" ;

float LC_velocity(DCS, time) ;

LC_velocity:_FillValue = -999.f ;

LC_velocity:units = "m/s" ;

LC_velocity:long_name = "instantaneous velocity" ;

float LC_semiminor_235K(DCS, time) ;

LC_semiminor_235K:_FillValue = -999.f ;

LC_semiminor_235K:units = "km" ;

LC_semiminor_235K:long_name = "Semi-minor axis of the equivalent ellipse at a 235K threshold" ;

float LC_semimajor_235K(DCS, time) ;

LC_semimajor_235K:_FillValue = -999.f ;

LC_semimajor_235K:units = "km" ;

LC_semimajor_235K:long_name = "Semi-major axis of the equivalent ellipse at a 235K threshold" ;

float LC_ecc_235K(DCS, time) ;

LC_ecc_235K:_FillValue = -999.f ;

LC_ecc_235K:units = "semiminor/semimajor" ;

LC_ecc_235K:long_name = "Eccentricity of the equivalent ellipse at a 235K threshold" ;

float LC_orientation_235K(DCS, time) ;

LC_orientation_235K:_FillValue = -999.f ;

LC_orientation_235K:units = "degrees" ;

LC_orientation_235K:long_name = "orientation of the equivalent ellipse at a 235K threshold" ;

float LC_semiminor_220K(DCS, time) ;

LC_semiminor_220K:_FillValue = -999.f ;

LC_semiminor_220K:units = "km" ;

LC_semiminor_220K:long_name = "Semi-minor axis of the equivalent ellipse at a 220K threshold" ;

float LC_semimajor_220K(DCS, time) ;

LC_semimajor_220K:_FillValue = -999.f ;

LC_semimajor_220K:units = "km" ;

LC_semimajor_220K:long_name = "Semi-major axis of the equivalent ellipse at a 220K threshold" ;

815  float LC_ecc_220K(DCS, time) ;

LC_ecc_220K:_FillValue = -999.f ;

LC_ecc_220K:units = "semiminor/semimajor" ;

LC_ecc_220K:long_name = "Eccentricity of the equivalent ellipse at a 220K threshold" ;

float LC_orientation_220K(DCS, time) ;

820  LC_orientation_220K:_FillValue = -999.f ;

LC_orientation_220K:units = "degrees" ;

LC_orientation_220K:long_name = "orientation of the equivalent ellipse at a 220K threshold" ;

int LC_surfPix_235K(DCS, time) ;

LC_surfPix_235K:_FillValue = -999 ;

825  LC_surfPix_235K:units = "number of pixels" ;

LC_surfPix_235K:long_name = "Cold cloud surface in number of pixels of the convective cluster for a 235K threshold" ;

int LC_surfPix_210K(DCS, time) ;

LC_surfPix_210K:_FillValue = -999 ;

LC_surfPix_210K:units = "number of pixels" ;

830  LC_surfPix_210K:long_name = "Cold cloud surface in number of pixels of the convective cluster for a 210K threshold" ;

float LC_surfkm2_235K(DCS, time) ;

LC_surfkm2_235K:_FillValue = -999.f ;

LC_surfkm2_235K:units = "km2" ;

LC_surfkm2_235K:long_name = "Cold cloud surface in km2 of the convective cluster for a 235K threshold" ;

835  float LC_surfkm2_220K(DCS, time) ;

LC_surfkm2_220K:_FillValue = -999.f ;

LC_surfkm2_220K:units = "km2" ;

LC_surfkm2_220K:long_name = "Cold cloud surface in km2 of the convective cluster for a 220K threshold" ;

float LC_surfkm2_210K(DCS, time) ;

840    LC_surfkm2_210K:_FillValue = -999.f ;

    LC_surfkm2_210K:units = "km2" ;

    LC_surfkm2_210K:long_name = "Cold cloud surface in km2 of the convective cluster for a 210K threshold" ;

   float LC_surfkm2_200K(DCS, time) ;

    LC_surfkm2_200K:_FillValue = -999.f ;

845    LC_surfkm2_200K:units = "km2" ;

    LC_surfkm2_200K:long_name = "Cold cloud surface in km2 of the convective cluster for a 200K threshold" ;

 // global attributes:

    :title = "TOOCAN - Morphological characteristics of the Deep Convecive Systems initiating between 01/01/2012 00:00 UTC and 01/31/2012 23:30
850 UTC" ;

    :creator_name = "Thomas Fiolleau" ;

    :contributor_name = "Remy Roca" ;

    :contact = "thomas.fiolleau@cnrs.fr" ;

    :institution = "CNRS/LEGOS/IPSL" ;

855    :conventions = "CF-1.6, ACDD-1.3" ;

    :tracker = "TOOCAN" ;

    :version = "2.08" ;

    :Geostationary_platform = "MSG2" ;

    :region = "AFRICA" ;

860    :region_longitude = "-55 -  55" ;

    :region_latitude = "-40 -  40" ;

    :temporal_resolution = "30 min" ;

    :Spatial_resolution = "0.04 degree" ;

    :time_coverage_start = "01/01/2012 00:00 UTC" ;

865    :time_coverage_End = "02/01/2012 17:00 UTC" ;

    :DCS_occurence = "40915" ;

## Appendix B

**Table B1**. Integrated morphological parameters of each identified deep convective systems documented in the TOOCAN netCDF-4 and ASCII monthly tracking files.

| Integrated morphological parameters | Description | Units |
|---|---|---|
| DCS_number | Label of the DCS in the segmented images | / |
| INT_qltyDCS | Quality flag indicating if the DCS initiates or dissipates due to missing images | / |
| INT_classif | Classification of the DCS according to the Roca etal (2017) definition | / |
| INT_duration | Life time duration | hr |
| INT_UTC_timeInit | Universal Time of the DCS initiation | seconds since 01/01/1970 |
| INT_localtime_Init | Local time of the DCS initiation | seconds since 01/01/1970 |
| INT_lonInit | Longitude of the DCS center of mass at its initiation | degrees |
| INT_latInit | Latitude of the DCS center of mass at its initiation | degrees |
| INT_UTC_timeEnd | Coordinated Universal Time of the DCS dissipation | seconds since 01/01/1970 |
| INT_localtime_End | Local time of the DCS dissipation | seconds since 01/01/1970 |
| INT_lonEnd | Longitude of the DCS center of mass at its dissipation | degrees |
| INT_latEnd | Latitude of the DCS center of mass at its dissipation | degrees |
| INT_velocityAvg | Average velocity of the DCS from its initiation to its dissipation | m/s |
| INT_distance | Distance covered by the DCS | km |
| INT_lonmin | Minimum longitude of the DCS along its life cycle | degrees |
| INT_latmin | Minimum latitude of the DCS along its life cycle | degrees |
| INT_lonmax | Maximum longitude of the DCS along its life cycle | degrees |
| INT_latmax | Maximum latitude of the DCS along its life cycle | degrees |
| INT_TbMin | Minimum brightness temperature of the DCS along its life cycle | K |
| INT_surfmaxPix_235K | Maximum cloud surface reached by the DCS at 235K | number of pixels |
| INT_surfmaxkm2_235K | Maximum cloud surface reached by the DCS at 235K | km² |
| INT_surfmaxkm2_220K | Maximum cloud surface reached by the DCS at 220K | km² |
| INT_surfmaxkm2_210K | Maximum cloud surface reached by the DCS at 210K | km² |
| INT_surfmaxkm2_200K | Maximum cloud surface reached by the DCS at 200K | km² |
| INT_surfcumkm2_235K | DCS total cold cloudiness at 235K from its initiation to its dissipation | km² |
| INT_classif_JIRAK | DCS classification according to the Jirak etal (2003) definition | / |
| INT_classif_MADDOX | DCS classification according to the Maddox (1981) definition | / |
| INT_TSnumber_IBTRACS | number of the Tropical Storm in the IBTrACS file associated with the DCS in a 1000km radius | / |
| INT_TSnature_IBTRACS | nature of the Tropical Storm in the IBTrACS file | / |

| | | |
|---|---|---|
| INT_TSmindistance_IBTRACS | Distance of a DCS to a Tropical Storm (max: 1000km) | km |

**Table B2**. morphological parameters described every 30 minutes along the life cycles of each deep convective systems documented in the TOOCAN netCDF-4 and ASCII monthly tracking files.

| morphological parameters along the DCS life cycles | Description | Units |
|---|---|---|
| QCgeo_IRimage | Flag Indicating the IR missing image | |
| LC_tbmin | Minimum brightness temperature | K |
| LC_tbavg_235K | Average brightness temperature at 235K | K |
| LC_tbavg_208K | Average brightness temperature at 208K | K |
| LC_tbavg_200K | Average brightness temperature at 200K | K |
| LC_tb_90th | 90th percentile of brightness temperature | K |
| LC_UTC_time | Coordinated Universal Time of the DCS | seconds since 01/01/1970 |
| LC_localtime | Local time of the DCS | seconds since 01/01/1970 |
| LC_lon | Longitude of the center of mass | degrees |
| LC_lat | Latitude of the center of mass | degrees |
| LC_x | Column of the center of mass in the image | Indices of the column |
| LC_y | Line of the center of mass in the image | Indices of the line |
| LC_velocity | Instantaneous velocity | m/s |
| LC_sminor_235K | Semi-minor axis of the ellipse at a 235K threshold | km |
| LC_smajor_235K | Semi-major axis of the ellipse at a 235K threshold | km |
| LC_ecc_235K | Eccentricity of the ellipse at a 235K threshold computed as $\frac{LC\_sminor\_235K}{LC\_smajor\_235K}$ | / |
| LC_orientation_235K | Orientation of the ellipse at a 235K threshold | degrees |
| LC_sminor_220K | Semi-minor axis of the ellipse at a 220K threshold | km |
| LC_smajor_220K | Semi-major axis of the ellipse at a 220K threshold | km |
| LC_ecc _220K | Eccentricity of the ellipse for a 220K threshold computed as: $\frac{LC\_sminor\_220K}{LC\_smajor\_220K}$ | / |
| LC_orientation _220K | Orientation of the ellipse at a 220K threshold | degrees |
| LC_surfPix_235K | Cold cloud surface of the convective cluster for a 235K threshold | number of pixels |
| LC_surfPix_210K | Cold cloud surface of the convective cluster for a 210K threshold | number of pixels |
| LC_surfkm2_235K | Cold cloud surface of the convective cluster for a 235k threshold | km² |
| LC_surfkm2_220K | Cold cloud surface of the convective cluster for a 220k threshold | km² |
| LC_surfkm2_210K | Cold cloud surface of the convective cluster for a 210k threshold | km² |
| LC_surfkm2_200K | Cold cloud surface of the convective cluster for a 200k threshold | km² |

## Appendix C

**Table C1.** Morphological parameters of each DCS gridded into a 1° × 1° daily grid of the CACATOES netCDF-4 monthly files.

| Variable | Description | Units |
|---|---|---|
| DAILY_DCS_Cloudcover | Daily cloud cover | % |
| QCgeo_numgeo | ID of the geostationary platform | / |
| QCgeo_nbMissingImages | Number of missing/corrupted geo images per day | / |
| QCgeo_GEOScanMode | Number of GEO images per day | / |
| QCtoocan_Interruption | Indication of a tracking interruption | / |
| QCtoocan_nbSegmentedImages | Number of TOOCAN segmented images | / |
| QCtoocan_trackingOK_allplatforms | Quality of the Tracking over the entire tropical belt | / |
| QCcacatoes_nbpixels | Number of GEO pixels within a 1°/1day CACATOES gridpoint | / |
| QCcacatoes_SurfGridPoint | Cumulated GEO pixels surface into a CACATOES 1°/1day gridpoint | / |
| INT_DCSnumber | Label of the DCS | / |
| QC_DCS | Confidence on the tracked DCS | / |
| INT_classif | DCSs classification according to Roca etal (2017) definition | / |
| INT_duration | Life time duration | h |
| INT_surfmaxkm2_235K | Maximum cold cloud surface reached by the DCS along its life cycle at 235K | km² |
| INT_surfmaxkm2_220K | Maximum cold cloud surface reached by the DCS along its life cycle at 220K | km² |
| INT_surfmaxkm2_210K | Maximum cold cloud surface reached by the DCS along its life cycle at 210K | km² |
| INT_surfmaxkm2_200K | Maximum cold cloud surface reached by the DCS along its life cycle at 220K | km² |
| INT_surfcumkm2_235K | Cumulated cold cloud surface of the DCS along its life cycle at 235K | km² |
| INT_Tmax | Time of maximum extent at 235K | % |
| INT_SurfDCS_220K_at_Tmax | DCS size at 220K at time of maximum extent | km² |
| INT_SurfDCS_210K_at_Tmax | DCS size at 210K at time of maximum extent | km² |
| INT_SurfDCS_200K_at_Tmax | DCS size at 200K at time of maximum extent | km² |
| INT_Tbavg235K_at_Tmax | average Tb lower than 235K at time of maximum extent | K |
| INT_Tbavg208K_at_Tmax | average Tb lower than 208K at time of maximum extent | K |
| INT_Tbavg200K_at_Tmax | average Tb lower than 200K at time of maximum extent | K |
| INT_Tb90th_at_Tmax | 90th percentile of Tb at time of maximum extent | K |
| INT_Ecc220K_at_Tmax | eccentricity of the DCS at time of maximum extent for a 220K threshold computed as: $\frac{Sminor\_220K}{Smajor\_220K}$ | / |
| INT_Ecc235K_at_Tmax | eccentricity of the DCS at time of maximum extent for a 235K threshold computed as: $\frac{Sminor\_235K}{Smajor\_235K}$ | / |
| INT_orientation220K_at_Tmax | orientation of the DCS at time of maximum extent for a 220K threshold | degrees |
| INT_orientation235K_at_Tmax | orientation of the DCS at time of maximum extent for a 235K threshold | degrees |
| INT_Distance | Propagated distance covered by the DCS | km |
| INT_Tbmin | Minimum brightness temperature of the DCS along its life cycle | K |
| INT_SurfDCS_235K | DCS Integrated Surface at a 235K threshold within a CACATOES 1°/1day grid point | km² |
| INT_SurfDCS_220K | DCS Integrated Surface at a 220K threshold within a CACATOES 1°/1day grid point | km² |
| INT_SurfDCS_210K | DCS Integrated Surface at a 210K threshold within a CACATOES 1°/1day grid point | km² |

| | | |
|---|---|---|
| INT_SurfDCS_200K | DCS Integrated Surface at a 200K threshold within a CACATOES 1°/1day grid point | km² |
| INT_GridFraction_235K | Fraction of the CACATOES 1°/1day grid point occupied by a DCS at a 235K threshold | % |
| INT_GridFraction_220K | Fraction of the CACATOES 1°/1day grid point occupied by a DCS at a 220K threshold | % |
| INT_GridFraction_210K | Fraction of the CACATOES 1°/1day grid point occupied by a DCS at a 210K threshold | % |
| INT_GridFraction_200K | Fraction of the CACATOES 1°/1day grid point occupied by a DCS at a 200K threshold | % |
| INT_gridtimeOccupation_start | start time of the grid point occupation by a DCS | hr since the start of the day |
| INT_gridtimeOccupation_end | End time of the grid point occupation by a DCS | hr since the start of the day |
| INT_Sfract_235k | Fraction of the DCS within the CACATOES 1°/1day grid point at a 235K threshold | % |
| INT_Sfract_220K | Fraction of the DCS within the CACATOES 1°/1day grid point at a 220K threshold | % |
| INT_Sfract_210K | Fraction of the DCS within the CACATOES 1°/1day grid point at a 210K threshold | % |
| INT_Sfract_200K | Fraction of the DCS within the CACATOES 1°/1day grid point at a 220K threshold | % |
| INT_TSnature_IBTRACS | nature of the Tropical Storm in the IBTrACS file | / |
| INT_TSnumber_IBTRACS | number of the Tropical Storm in the IBTrACS file associated with the DCS in a 1000km radius | / |
| INT_TSmindistance_IBTRACS | Distance of DCS to the IBTrACS Tropical Storm | Km |
| INT_classif_JIRAK | DCS classification according to the Jirak etal (2003) definition | / |
| INT_classif_MADDOX | DCS classification according to the Maddox (1980) definition | / |
| INIT_Time | time of the DCS initiation | seconds since 01/01/1970 |
| INIT_Lon | Longitude of the DCS center of mass at its initiation | degrees |
| INIT_Lat | Latitude of the DCS center of mass at its initiation | degrees |
| END_Time | time of the DCS dissipation | seconds since 01/01/1970 |
| END_Lon | Longitude of the DCS center of mass at its dissipation | degrees |
| END_Lat | Latitude of the DCS center of mass at its dissipation | degrees |

## Video supplement

Video S1 (https://doi.org/10.5446/68200, Fiolleau 2024) shows an animation of convective situation segmented by TOOCAN from HIMAWARI IR data over the Western Pacific region in October 2015.

## Author contributions

Thomas Fiolleau and Remy Roca initiated the work. Thomas Fiolleau and Remy Roca prepared some datasets. Thomas Fiolleau drafted the figures. All the authors contributed to the writing of the paper.

**Competing interests**

The authors declare that they have no conflict of interest.

**Acknowledgements**

This work has been supported by CNRS and CNES under the Megha-Tropiques program. We thank the ESPRI/IPSL team to providing computing and storage resources. We thank the French data center AERIS, and especially Sophie Cloché to help accessing the data (georing, ScaRaB). The authors thank also L. Gouttesoulard for his help handling the geostationary satellite data archive. The authors further acknowledge M. Dejus, project manager at CNES and his team for their help on the technical 895 aspects of ScaRaB. They thank also Nuria Duran-Gomez and Rémi Jugier from MAGELLIUM for their contribution on the GEOring homogenization.

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
