# Peer review of "A Deep Convective Systems Database Derived from the Intercalibrated Meteorological Geostationary Satellite Fleet and the TOOCAN algorithm (2012-2020)"

_Earth System Science Data, 2024_

## Author Comment (AC1)

Dear Editor,

We would like to thank the reviewers for their comments. The reviewers have made a number of comments that will benefit this manuscript. In response to their thorough evaluations, we have carefully revised our manuscript to address each question and concern. Detailed responses to their comments can be found in the author comments.

We have also adjusted the manuscript accordingly to the journal standards:

- The reference list has been compiled according to the journal standards.
- We have adjusted the color schemes in the figures 1 and 2 by using a Color Universal Design (CUD) palette, to ensure that colors are distinguishable by everyone, combined with some Different line styles and markers to enhance distinguishability. All the figures have been checked using the Coblis–Color Blindness Simulator to check for distinguishability. Figs. 1, 2, 7 have passed the test. More specifically, we have determined that Fig. 7 is sufficiently distinguishable for individuals with color blindness without any modifications.
- Finally, the color cells of the Tables have been removed to fit with the journal requirements.

As supplement to the paper, a video (https://doi.org/10.5446/68200, Fiolleau 2024) has been added to the paper to illustrate at a 30-minute time frequency, showing the DCS identified by TOOCAN in October 2015 over the Western Pacific region.

---

## Author Comment (AC2)

**Reviewer #1**

Title: A Deep Convective Systems Database Derived from the Intercalibrated Meteorological Geostationary Satellite Fleet and the TOOCAN algorithm (2012-2020)

Author(s): Thomas Fiolleau and Remy Roca

MS No.: essd-2024-36

MS type: Data description paper[

Review Comments: General Comment.

This manuscript describes the database derived from the intercalibrated geostationary satellite constellation (GEOring), processed by the TOOCAN algorithm to produce a historical dataset of the trajectories and characteristics of the mesoscale convective system over the 9 years from 2012 to 2020. The dataset is composed of two databases, the TOOCAN, describing the trajectories, including the radiative and morphological characteristics of the MCS with a 30-minute time step, and the CACATOES, a 1°x1° global monthly file describing the spatial distribution of the DCS density over the tropical belt. This is a very rich dataset and the manuscript describes in detail the procedures used, presents some results and shows some examples. I consider the manuscript suitable for publication in the ESSD after some minor corrections.

*We would like to thank the reviewer for the positive comments and appreciations. The reviewer #1 has made a number of comments that will benefit this manuscript. We have thoroughly revised our manuscript and addressed all the questions and concerns raised by the reviewer #1. Detailed responses to their comments are provided in the following. the original comments of the reviewers are in black and our responses in blue and italic.*

*We have also adjusted the manuscript accordingly to the journal standards:*

- *The reference list has been compiled according to the journal standards.*
- *We have adjusted the color schemes in the figures 1 and 2 by using a Color Universal Design (CUD) palette, to ensure that colors are distinguishable by everyone, combined with some Different line styles and markers to enhance distinguishability. All the figures have been checked using the Coblis–Color Blindness Simulator to check for distinguishability. Figs. 1, 2, 7 have passed the test. More specifically, we have determined that Fig. 7 is sufficiently distinguishable for individuals with color blindness without any modifications.*
- *Finally, the color cells of the Tables have been removed to fit with the journal requirements.*

*As supplement to the paper, a video ([https://doi.org/10.5446/68200](https://doi.org/10.5446/68200), Fiolleau 2024) has been added to the paper to illustrate at a 30-minute time frequency, showing the DCS identified by TOOCAN in October 2015 over the Western Pacific region.*

```
Specific Comments:
```

1) Line 40 – Rewrite the sentence – difficult to understand the meaning, as well as the verb agreement.

*Thank you for your comment. We have improved the sentence for clarity.*

*Despite a long research history (Houze, 2018), understanding the lifetime duration of tropical convective systems and the lengths of their various phases throughout their life cycles in the current climate, as well as their evolution in a warmer and moister world, remains challenging (Roca et al., 2020).*

2) Line 43 – "MCS for a family of deep convective systems with horizontal scales larger than 100 km". When you mention 100 km horizontal scale, it corresponds to an MCS of roughly with 10000 km². However, you start to look MCS from an area larger than 635 km². Please clarify and justify the choice.

*To identify individual deep convective systems (DCS) through their life cycles, the TOOCAN algorithm relies an iterative process of detection and dilatation of convective seeds in the spatio-temporal domain to identify convective systems. The iterative process consists in a multi-thresholding of the convective seeds ranging from 190K to 235K, with a 2K brightness temperature increment. At each brightness temperature threshold, we identify convective seeds respecting a minimum lifetime duration criterion of 3 frames (1h30) and an area criteria of 625 km² per frame. Each brightness temperature threshold is followed by a dilation in 3 dimensions of the identified convective seeds, if any, which involves aggregating the neighboring pixels spatially and temporally to their respective convective seeds.*

*At the end of the process, the maximum surfaces reached by the individual DCS throughout their life cycles are larger than the initial area criterion of 625 km². Figure 9 shows that the maximum extent of identified DCS with such a technique ranges from approximately 1,000 km² to several million km².*

*However, we want to point out that we differentiate Deep Convective Systems (DCS) from Mesoscale Convective Systems (MCS). According to the description given by Houze (1993), an MCS is a convective system producing contiguous precipitation on horizontal scales larger than 100 km. In the introduction, we aimed to clarify that not all Deep Convective Systems reach this 100 km horizontal scale, meaning that MCS form a specific category within the broader group of Deep Convective Systems.*

*We have reformulated the sentence for clarity:*

*Within this full spectrum of organized convection, Mesoscale convective systems (MCS) form a specific family of deep convective systems producing contiguous precipitation on horizontal scales larger than 100 km (Houze 1993) and which can organize into mesoscale convective complexes (Maddox 1980), squall-line systems, super cluster (Mapes and Houze 1993).*

3) Line 47-50. Please rewrite for better clarity.

*We have simplified our text to make clearer.*

4) Line 53 – Infirm?

*Thank you for your comment.*

*We replaced « infirm » by « refine ». Here is the corrected sentence:*

*Line 53 : "In particular, after decades of field campaigns and detailed case studies, satellite climatology can now support the statistical analysis needed to refine our current understanding."*

```
5) Line 57 – Please correct - algorithms…
   Line 57 – The justification is not appropriate now, in
   nowadays, there are the ISCCP H series, the MERGIR, among
   others.
```

*Thanks for your comment. We have suppressed the sentence for more clarity. The global IR datasets as well as the cloud tracking datasets are already presented Table1 and discussed later in the paragraph.*

```
6) Line 62 – "A deep convective system then corresponds to a
   succession of convective clusters " Should be in the other way
   round ?, and why only deep convective, stratiform clouds also
   is part of MCS.
```

*Thank you for your comment. However, the definition of deep convective systems has been done previously in the text line 46:*

*"Whatever the scale and degree of organization, DCS are composed of a convective core where heavy rainfall takes place, associated to a stratiform anvil with lighter precipitation as well as nonprecipitating cirriform cloudiness. DCS are further characterized by a life cycle. DCS initiate and develop from one or more individual deep convective cells, which organize themselves into convective cores. The system then enters a maturity phase in which a stratiform part and cirriform cloud develop sustained by older convective cells. Finally, as convection vanishes, DCS are no longer fed by convective cells, they dissipate and scatter into several individual cirriform clouds."*

*Based on this definition, we explain how the main tracking algorithms work to detect and track DCS. For more clarity and to define the term 'cloud cluster' (see comment #8), we have modified this part of the text at line 60. We have retained the formulation: 'A deep convective system through its life cycle corresponds to a succession of cloud clusters in a time sequence of IR images,' as it effectively explains how convective systems are handled using time sequences of IR geostationary imagery.*

*Line 60: "Tracking algorithms are generally based on two steps: a detection step to identify contiguous areas of cold temperatures in a single IR image, referred to as cloud clusters in which deep convection is organized, and a tracking step to match the identified cloud clusters from one frame to the next. A deep convective system through its life cycle then corresponds to a succession of cloud clusters in a time sequence of IR images."*

```
7) Line 85 – There is a new paper (Km-Scale Simulations of
   Mesoscale Convective Systems Over South America—A Feature
   Tracker Intercomparison - 10.1029/2023JD040254). This is study
   is very complementary to the discussions on this manuscript.
```

*The paper of Prein etal (2024) has already been cited line 103 and 109. We have updated the reference as the paper of Prein etal (2024) has been accepted in April 2024.*

*Prein, A. F., and Coauthors, 2024: Km-Scale Simulations of Mesoscale Convective Systems Over South America—A Feature Tracker Intercomparison. Journal of Geophysical Research: Atmospheres, **129**, https://doi.org/10.1029/2023JD040254.*

8) Line 88 – I think it is important to define what the author calls a cluster of pixels. Later in the text I noticed that you are looking at the 4 neighbouring pixels and the diagonal pixels. It should be clearly defined what you are tracking.

*Thank you for you comment.*

*However, we talked more about 'cloud cluster' than 'cluster of pixels'. Cloud cluster refers to contiguous areas of cold temperatures in infrared (IR) images where deep convection is organized. These cloud clusters are tracked over time to study the lifecycle of deep convective systems.*

*We changed the sentence line 60 to clarify the term of "cloud cluster", as explained in the comment #6.*

*We have also made changes in the text, using 'cloud cluster' whenever necessary.*

*Regarding your comment on the 4 neighboring pixels and the diagonal pixels, it refers to the dilation step of the TOOCAN algorithm. Dilation is a morphological operation in image processing used to expand the boundaries of an object in a segmented image. It relies on a kernel operator, which defines the neighborhood around each pixel where the dilation operation will be applied. In the case of TOOCAN algorithm, we use a 10-connected spatiotemporal neighborhood kernel operator, composed of an 8-connected spatial neighborhood and a 2-connected temporal neighborhood, to favor spatial spread over temporal spread.*

*The TOOCAN algorithm has been fully described in Fiolleau and Roca (2013). However, we modified part of the text line 274 for more clarity about the spatiotemporal neighborhood kernel operator and the functioning of TOOCAN algorithm:*

*"This dilation step involves adding pixels belonging to the intermediate cold cloud shield to all previously detected convective seeds using a 10-connected spatiotemporal neighborhood kernel operator, composed of an 8-connected spatial neighborhood and a 2-connected temporal neighborhood, to favor spatial spread over temporal spread."*

9) I also suggest talking about the RDT algorithm, which also uses a variable threshold to follow the MCS structure.

What are the basic differences between the two methods?

Also, I see some similarities of the TOOCAN with the Fortracc, is the TOOCAN based in this tracking with the variable threshold?

*Thank you for tour comment.*

*RDT has been developed in support to nowcasting perspectives, with the objective to detect rapid developing convective cells. By applying adaptative BT thresholds on a single IR image, the initiation of individual convective cells is detected as early as possible. Each cell is then detected by a specific BT*

*threshold. Then the tracking step is based on a geographical overlapping of "cells" between two successive infrared images.*

*I have added the following reference of the RDT algorithm in the introduction section:*

*Autones, F., and J.-M. Moisselin, 2013: Algorithm Theoretical Basis Document for "Rapid Development Thunderstorms" (RDT-PGE11 v3.0), Scientific documentation of SAF/NWC, code SAF/NWC/CDOP2/MFT/SCI/ATBD/11. http://www.nwcsaf.org/ (Accessed June 10, 2024).*

*FORTRACC is an algorithm for tracking and forecasting morphological characteristics of MCSs through their entire life cycles from IR observation of geostationary. The cloud tracking algorithm is similar to the methodology introduced in Williams and Houze (1987) and Mathon and Laurent (2001). It is based on two steps: a cloud cluster detection method based on a threshold temperature (235 K) applied on single IR images which identify cold cloud shield. The tracking technique is based on area overlapping of cold cloud shields in successive images (Vila etal 2008). A forecast module has been added to the traditional detection and tracking steps to nowcast the evolution of each individual MCS.*

*The TOOCAN algorithm then widely differs technically from the RDT and FORTRACC algorithm, in that way that it detects and tracks individual convective systems through their life cycles not anymore with the traditional 2 steps described previously but in a single step within a space and time volume of IR images. A region growing image processing technique is applied in this spatio-temporal volume consisting in an iterative process of several detections and dilatation of 3D convective seeds from 190K BT threshold to 235K threshold. The iterative process is stopped when the 235K threshold is reached, corresponding to the boundaries of the high cold cloud shield. I agree that the 235K threshold for identifying the boundaries of the high cloud is identical to that used by FORTRACC algorithm. However, while FORTRACC focuses only on identifying high cold clouds using a single threshold at 235K, TOOCAN, through its region-growing method, goes further by decomposing these high cold clouds identified with the same BT threshold into different deep convective systems.*

```
10)    Line 113 - Sometime the split is a natural characteristic
    of the MCS, as for instance the supercells that regularly split
    in two different systems.
```

*Merging and splitting of convective cells is common (Bluestein et al., 1990)  and is an important process in convective systems dynamics in the mid-latitudes. Convective systems grow upscale from multiple individual cells and can decay into multiple individual cirriform clouds as they weaken (Ćurić et al., 2009; Rotunno et al., 1988). As well, at a larger scale, the high cold cloud shields can be shared between several DCSs, whose anvil clouds can split and merge with each other over time.*

*The objective of TOOCAN is to associate these individual merging and splitting cells with a single convective system, and at the dissipation stage, to associate the scattered cirriform clouds with the same convective system, ensuring that its life cycle is fully resolved.*

*For the case of a supercell splitting in two systems, if the maximum of convection (minimum of BT) of these 2 systems occurs after the 'split' time of the supercell, the TOOCAN will segment 2 DCS. In the case the maximum of convection occurs before the split, only one DCS will be identified.*

11)     Line 120 – "MCS shield growth rate has been carried out (Elsaesser et al. 2022)" . Several other papers earlier discussed the growth rate, one example is the Machado and Laurent 2004 ([https://doi.org/10.1175/1520-0493(2004)132<0714:TCSAEO>2.0.CO;2](https://doi.org/10.1175/1520-0493(2004)132<0714:TCSAEO>2.0.CO;2).).

*Thank you for your comment.*

*You are right. Machado and Laurent (2004) discuss the growth rate of the MCS cold cloudshield. However, the study carried by Machado and Laurent (2004) is limited to the Amazonian region. Here, we aimed to illustrate the fact that deep convective systems databases at tropical scale have been used to improve our understanding of tropical convection.*

*We have changed the sentence accordingly:*

*"For instance, Elsaesser et al. (2022) investigated the growth rate of the cloud shield across the entire tropical belt."*

12)     Table I – correct Feng etal.

*We have corrected the citation (Feng et al. 2021).*

13)     Page 145 – "This paper presents a such DCS database" – rewrite the sentence.

*Thank you. We have corrected the error.*

14)     Page 185 – "The decrease of the ScaRaB data availability does not impact the quality of the corrections from the end of 2018". If there is no effect, why not apply the correction once a month to have a more homogeneous dataset?

*The inter-calibration of infrared observations from geostationary satellites using the ScaRaB data as the reference instrument was carried out from 2012 to 2016 and published in 2020. The inter-calibration coefficients were calculated every 10 days. This period was chosen to ensure statistical robustness while preventing the inclusion of higher frequency variations (less than 10 days) caused by calibration issues, such as those due to decontamination procedures. Indeed, the accumulation of ice on the surface of the imager optics modifies their spectral response function and consequently affects the measurement of the temperatures. To face these issues, some decontamination events are regularly applied on the radiometers inducing instability in the temperature measurements. This effect has been particularly shown for the MVIRI instrument on board the METEOESAT-7 geostationary platform.*

*For the extension of the database up to 2020, we faced the challenge of the decrease of the ScaRaB's availability from 2018, making it difficult to compute inter-calibration coefficients. As explained in the paper, to overcome the lack of ScaRaB observations, the calculation of inter-calibration and spectral normalization coefficients was performed every 30 days instead of the initial 10 days. We conducted an analysis, presented in Figure 1, to examine the impact of moving from 10 days to 30 days. First, METEOSAT-7, which was greatly impacted by calibration issues, was replaced in 2017 by MSG-1, which*

*has fewer issues. Moreover, the results show no impact on the residuals of the regressions, indicating that the IR GEOring database can be considered homogeneous throughout the 2012-2020 period.*

*T. J. Hewison et al., "GSICS inter-calibration of infrared channels of geostationary imagers using metop/IASI," IEEE Trans. Geosci. Remote Sens., vol. 51, no. 3, pp. 1160–1170, 2013.*

15)      Page 188 – Authors should discuss the larger Bias of
         Meteosat-7.

*Thank you for your comment.*

*The larger bias of MET-7 has been fully discussed in Hewison et al (2013) and in Fiolleau et al 2020. We added in the text a sentence discussing the calibration issue of the IR channel of METEOSAT-7 with a reference to the Hewison et al (2013) and Fiolleau et al. (2020) papers.*

*"The large bias observed in the MET7/MVIRI calibration between 2012 and 2017 (Figure 1a) is explained by significant ice contamination on the MVIRI optics, as fully discussed in Hewison et al. (2013) and Fiolleau et al. (2020). »*

16)      Page 197 – How does the interpolation take into account
         differences in satellite resolution?

         How is the change in resolution with latitude zenith angle
         taken into account? How do you deal with missing data, are they
         extrapolated?

         How does the different resolution affect the size of the MCS?

         Is the 55N to 55S window too large for an acceptable satellite
         viewing angle?

*All the IR geostationary data have been remapped into a common longitude-latitude 0.04° equal-angle grid as explained line 197 and and the methodology has been fully detailed in Fiolleau et al. (2020). The regridding process is performed by applying the inverse distance weighting method in the radiance space. The radiance value of each given 0.04° pixel is then computed by averaging the radiance values of its neighborhood pixels. Then, the average radiance is transformed in brightness temperature using the Planck function in order to account for its the non-linearity.*

*The pixels of a longitude-latitude equal-angle grid have consistent angular distances between them as the zenith angle increases. However, the spatial resolution of the pixel varies with latitude from ~19.75 km² at nadir to ~15.2km² and ~11.40km² for a 40° and a 55° latitude respectively. This variation in spatial resolution with increasing latitude has been accounted for in our processing.*

*As well, the missing pixels are not extrapolated.*

*The TOOCAN algorithm relies on an iterative process of detection and dilatation of convective seeds in the spatio-temporal domain to identify convective systems. The multi-thresholding of the convective seeds ranges from 190K to 235K, with a 2K brightness temperature increment. At each brightness temperature threshold, we identify convective seeds respecting a minimum lifetime duration of 3 frames (1h30) and exceeding 625 km² per frame. For a 0° latitude, 625km² per frame means around 31*

*pixels per frame, and for a 40° latitude, 41 pixels. First, the number of pixels required to detect convective seeds is relatively low at nadir and does not differ so much at a 40° latitude. Secondly, a convective seed that does not meet the criteria for duration or size at a given brightness temperature (BT) threshold could still meet them at a warmer BT threshold. For instance, a convective seed located at 40° latitude with an area lower than 41 pixels at a 200K threshold will not be selected, but it will be at a 202K threshold if its surface area exceeds 41 pixels. Finally, the 3D detection step is followed by a dilation step in the spatio-temporal domain to associate the convective seeds with their stratiform and cirriform clouds. As a result of this iterative process, the impact of varying spatial resolution with latitude on the size of the deep convective system is not significant.*

*A large zenith angle is not an issue for a convective system tracking purpose, as long as the brightness temperatures are corrected from the limb darkening effects. In this paper, we focused on building a deep convective system database over the tropics. Indeed, the lower tropopause altitude in mid-latitudes requires the adjustment of the convective seed detection thresholds applied in the TOOCAN algorithm. Moreover, a specific work has to be done to resolve the classification ambiguity for winter conditions in mid-latitudes, where meteorological fronts and convective systems often coexist.*

17)     `Figure 2 – "…in the range [180 K-235 K] between 2012 and 2020" why the range is different from the SCARAB correction [180 K-240 K]`

*Thank you for your comment.*

*Figures 1 and 2 show the results of the inter-calibration, spectral adjustment and limb correction of the GEOring IR data over an extended period from 2012 to 2020 compared with the period used for the Fiolleau et al (2020) paper. This homogenization method and analysis have been already detailed in Folleau etal (2020) for the 2012-2016 period.*

*The range 180K-240K corresponds to the range on which linear regressions are computed to calibrate and spectrally adjust the geostationary IR observation. Figure 1 shows the result of such calibration in the range [180K-240K].*

*Figure 2 shows the final evaluation step of the homogenization procedure, following the inter-calibration/spectral adjustment and limb darkening correction. In this validation step, we focused on the 180K-235K range, as the high cold cloud shield is identified in the TOOCAN algorithm by applying a 235K BT threshold on IR images.*

*For clarity, we have modified parts of the text:*

*Line 165: "This homogenization procedure includes the computation of the inter-calibration and spectral normalization coefficients every 10 days as well as the correction of the limb darkening effects impacting the brightness temperature measurements in the range [180K-240K] from geostationary satellites for the 2012-2016 period."*

*Line 190: "Figure 2 shows the variation of the biases of $BT_{GEO}$ for values lower than 235K between pairs of geostationary satellites monitoring a same region according to the difference of their zenith angles (ΔVZA), before and after zenith angle corrections. Without any corrections, $BT_{GEO}$\* bias varies from -5 K to +5 K as the ΔVZA moves from -50° to 50° regardless the geostationary platform (Figure 2a). By applying the zenith angle correction, this bias averages 0.21 K with a standard deviation of 1.33 K throughout the GEOring, independent of the variation in ΔVZA."*

18)     Line 225 – "The arrivals of GOES-16" – replace by launched;

*Thank you for your comment.*

*We changed the sentence as followed:*

*Line 228: The deployments of GOES-16 in 2018 and GOES-17 in December 2019 to their respective nadir positions over respectively the American and Eastern Pacific regions highly improve the observation of the southern regions.*

19)     Line 250 - Could you explain why the cloud cluster associated with the tropical cyclone was dropped from the track? Several times the MCS evolved from a tropical storm to a hurricane. In this case, does TOOCAN consider the MCS dissipated?

*Thank you for the comment. I am sorry for the confusion. The DCS are not dropped from the dataset, but flagged so that the user can or not filter out the DCS belonging to a given cyclone.*

*We revised the sentence as follow:*

*Line 237: To this end, we use the IBTrACS dataset to flag DCS belonging to a cyclonic circulation or classified themselves as tropical cyclones, allowing to either filter them out of the analysis or include them as needed.*

20)     Line 278 - There is an almost linear relationship between size (the threshold) and lifecycle. Your methodology uses the 1.5-hour lifecycle for the different thresholds. For 190K, 1.5 hours is a high requirement, but for 235K it is common.

*Thank you for your comment.*

*The impact of various brightness temperature thresholds on the area and lifetime of cloud clusters has been investigated in the paper describing the TOOCAN algorithm (Fiolleau and Roca, 2013). The results showed a small dependence of the cloud cluster sizes and lifetime duration to the temperature thresholds in the range 200 K–235 K. In this study, 50% of the cloud clusters segmented by thresholds within the range 200 K–235 K were due to systems with lifetime shorter than five frames (2h30). However, for a threshold of 190 K, cloud clusters lasting less than three frames (1h30) were indeed fewer in number but accounted for more than 50% of the population identified at this threshold. No cloud cluster segmented by a 190-K threshold lasted more than 12 frames (6h).*

*Concerning the TOOCAN methodology itself, a given brightness temperature threshold associated with a duration criterion (1h30) and a size criterion (625 km²) allows the detection of the individual elementary convective seed that will form an individual deep convective system at the end of the iterative process. A convective seed that does not meet the criteria for duration or size at a given Tb threshold could still meet them at a warmer brightness temperature threshold.*

21)     Line 280 - How do you deal with semi-transparent clouds? For pixels where the Tir does not correspond exactly to the

```
cloud top, the 3D threshold procedure will not work correctly
because it contradicts the basic principle of the 3D threshold?
Wouldn't it be better to follow the corrected cloud top?
```

*The upper limit of the high could cloud shield is identified in the TOOCAN algorithm by applying a 235K threshold on the IR imagery of geostationary satellites. This value of brightness temperature threshold has been commonly used in the literature (e.g., Kondo et al. 2006; Yuter and Houze 1998; Pope et al. 2008). In Bouniol et al. (2016), we have shown that such a threshold value of 235K represents a good compromise for attributing nonprecipitating clouds to a convective system, and to exclude semi-transparent clouds as well as mid- or low-level clouds that may develop under thinner cirrus clouds. We have also shown in Bouniol etal (2006) that for such a 235K brightness temperature threshold, the cloud-top altitude observed by the Cloudsat radar, is in the expected range for tropical cirrus.*

*Hence, the TOOCAN algorithm, relying on the 3D region growing technique with multi-thresholds applied in the IR geostationary observation can't be impacted by such an issue.*

- *Bouniol, D., R. Roca, T. Fiolleau, E. Poan, 2016: Macrophysical, microphysical and radiative properties of tropical Mesoscale Convective System along their life cycle. J. Climate, 29(9), 3353-3371. DOI: 10.1175/JCLI-D-15-0551.1*
- *Kondo, Y., A. Higuchi, and K. Nakamura, 2006: Small-scale cloud activity over the Maritime Continent and the western Pacific as revealed by satellite data. Mon. Wea. Rev., 134, 1581–1599, doi:10.1175/MWR3132.1.*
- *Pope, M., C. Jakob, and M. J. Reeder, 2008: Convective systems of the north Australian monsoon. J. Climte, 21, 5091–5112, doi:10.1175/2008JCLI2304.1.*
- *Yuter, S. E., and R. A. Houze Jr., 1998: The natural variability of precipitating clouds over the western Pacific warm pool. Quart. J. Roy. Meteor. Soc., 124, 53–99, doi:10.1002/qj.49712454504.*

22)     Line 282 – "to resolve the unphysical split and merge
        issues." There are split and merge in real life – Explain what
        do you consider unphysical?

*As explained in comment #10, merging and splitting of convective cells is common (Bluestein et al., 1990)  and is an important process in convective systems dynamics in the mid-latitudes. Convective systems grow upscale from multiple individual cells and can decay into multiple individual cirriform clouds as they weaken (Ćurić et al., 2009; Rotunno et al., 1988). As well, at a larger scale, the high cold cloud shields can be shared between several DCSs, whose anvil clouds can split and merge with each other over time.*

*The objective of TOOCAN is to associate these individual merging and splitting cells with a single convective system, and at the dissipation stage, to associate the scattered cirriform clouds with the same convective system, ensuring that its life cycle is fully resolved.*

*Here, the term 'unphysical' was not appropriate in the sentence, which has been reworded. We refer now to 'split and merge artifact':*

*Line 285: "Thanks to its spatio-temporal region-growing technique, the TOOCAN algorithm can track DCS by suppressing split and merge artifacts throughout their life cycles, which are inherent to classic overlap-based tracking techniques."*

23)      Line 287 – How the algorithm handles missing images. With overlapping, you would normally have to use image extrapolation or a dynamic overlap threshold. How does TOOCAN use the overlap threshold and what value is used?

*TOOCAN does not use any dynamic overlap or image extrapolation to continue the cloud tracking process in the presence of successive missing images. However, a time period up to 3 hours of consecutive missing images is filled halfway by the available IR image immediately preceding the gap and halfway by the available IR image following the gap. TOOCAN is then operated nominally on this time series of duplicated IR images.*

*We have detailed on how the algorithm handles missing data in comment #25.*

24)      Line 304 - Maybe I didn't understand correctly, but this is not a good example to show, because the advanced feature of TOOCAN is the 3D threshold to avoid splits and merges. However, looking at system A, it seems to be a unique MCS, but TOOCAN produced several MCS that merge (you said there was no merge) at 1200. Also, B appears to be a single MCS, but it changes to C. Could you provide a better explanation for these merges and splits we see in this image? It seems that TOOCAN solves many split and merge situations, but it continues to have split and merge in smaller numbers.

*Figure 4 illustrates the TOOCAN segmentations for a convective situation that occurred in October 2015 over the Western Pacific region, and aimed to demonstrate that the TOOCAN algorithm can identify the full spectrum of convective organization, from small and short-lived DCS to large and long-lived ones.*

*Recall that this figure shows snapshots of TOOCAN outputs every 6 hours, which does not facilitate precise and detailed observation and comments on the splits and merges of the systems, especially in such an extended region. Hence, with this illustration, we don't think it is possible to affirm that DCS A merged at 1200 UTC, nor that DCS B changed to C due to a merge or if this is a situation where the system B dissipates followed by an initiation of the system C (which is actually the case).*

*Moreover, the way the TOOCAN algorithm addresses split and merge issues has already been published and discussed in Fiolleau and Roca (2013), with various case studies over different regions of interest. Additionally, studies on the TOOCAN DCS life cycles over the past 10 years have demonstrated the algorithm's capability to handle split and merge situations (Roca etal 2017, Roca and Fiolleau 2020, Bouniol etal 2016, Bouniol etal 2020, Elsaesser etal 2022…).*

*To help the reviewer and the readers to better grasp this effect in TOOCAN, we have added supplementary material for this paper, including a Video S1 (https://doi.org/10.5446/68200, Fiolleau 2024) at a 30-minute time frequency, showing the DCS identified by TOOCAN in October 2015 over the Western Pacific region.*

25)     Figure 7 - I did not understand how a missing image could increase the bias in the number population of short lived MCS. It is also interesting to see that the 2 hour missing image has the largest bias and 4 hours no bias, for short lived MCS. I fully understand this for systems larger than about 8 hours, but for shorter lifetimes I don't.

Line 365 – Perhaps it was in this part of the manuscript that I understood the above problem. Actually, this is not a missing image, it is the replication of the same image. So it is not really a missing image, it represents that the MCS was frozen. Please clarify.

*A time period up to 3 hours of consecutive missing images is filled halfway by the available IR image immediately preceding the gap and halfway by the available IR image following the gap.*

*Several scenarios arise for the DCS facing such a time period of consecutive missing images:*

- *Some DCS that were supposed to initiate during this period are either detected at the time of resumption, in which case their duration is artificially shortened, or they cannot be identified at all.*
- *Some DCS that were supposed to dissipate during this period of missing images dissipate at the time of resumption, in which case their lifetime duration is artificially extended.*
- *Some DCS that initiate before the series of missing images and are expected to persist afterwards can still be tracked despite this interruption if they have sufficiently large cloud cover and if the convective situation evolves little during this gap.*
- *Some DCS can't overpass the gap of missing data and then artificially dissipate. Smaller systems may not survive the gap, resulting in shortened lifetimes, or their lifetimes may be artificially prolonged due to data replication.*
- *The diurnal cycle of DCS initiation also plays a role, with missing data between 2 PM and 6 PM on continental regions having a greater impact due to peak initiation compared to between midnight and 5 AM as we answer the comment #14 of the Reviewer #2.*

*The biases in the population of short-lived DCS, regarding the period of consecutive missing data, are then challenging to explain due to various these contributing factors.*

*Therefore, elucidating differences in observed biases for small systems proves relatively difficult. The 3-hour consecutive missing image period was primarily chosen to ensure continuity in tracking large systems, particularly during eclipse periods affecting METEOSAT-7 observations in India, which occur late in the day in August-September and February-March.*

*Conversely, the results regarding population bias for long-lived systems are more comprehensible and appear robust up to a 3-hour consecutive missing data period. This suggests that DCS can still be monitored despite the interruption if they possess sufficiently extensive cloud cover and if the convective situation evolves minimally during the gap.*

*We have made some modifications to Section 3.2.b to convey the information in a more detailed form.*

26)      Line 402 – "propagated distance, cold cloud maximum
    extent…)" – please do not use … and add all variables – maybe
    in a table – it is an important information to the users.

    Line 405 – "thresholds, eccentricity, instantaneous velocity…)"
    Same comment.

*We have added Appendix B to the paper, which describes all the variables available in the TOOCAN
database in Table B1 and Table B2. Additionally, Appendix C describes the variables in the CACATOES
database, as shown in Table C1.*

*We refer to these appendices and tables in the text.*

27)      Line 407 – ". A quality flag lower than 11110 is a good
    compromise to keep DCS little impacted by imagery issues".
    Please clarify what 11110 means.

*We have improved and clarified the paragraph describing the quality flag:*

*Line 423: "The first digit of the quality flag indicates whether a DCS is born naturally or due to
consecutive missing images, while the second digit indicates whether a DCS dissipates naturally or as a
result of consecutive missing images. The third digit reveals if a DCS is impacted by image boundaries
or missing pixels, with a value greater than one indicating such an issue.  Values of these three digits
greater than one indicate problems in initiation, dissipation, or touching the edge of the image. The
last two digits of the quality flag represent the number of missing images during the lifecycle of a given
DCS."*

28)      Figure 10 - It would be more appropriate to present this
    figure on the basis of the whole year, since not including the
    missing months, mainly during the dry season, will have an
    impact on the number of MCS/day..

*Thanks for the comment.*

*We agree that the map of the density of DCS can be impacted by the missing months in specific
regions, and specifically during the dry season. However, in this data paper we aim to introduce a
tropical database of Deep Convective Systems, which has been homogenized (calibration,
remapping…), but remains dependent to the varying amount of missing data across different regions,
as shown figure 3. In particular, the southern part of the Western pacific region and a band of the
southern Eastern Pacific region are not taken into consideration in our cloud tracking process because
they are monitored in a time frequency greater 30minutes by MTSAT from January 2012 to May 2015
and by GOES-15 from January 2012 to November 2017 respectively.*

*With this figure, we aimed first to show that the spatial distribution of the DCS population from our
database is consistent with the climatology of tropical convective systems, based on other definitions,
algorithms and satellite observations, either from local or global studies (Mohr and Zipser 1996, Liu et
al. 2008, Feng et al. 2021, Huang et al. 2018). At the same time, we also aimed to highlight to future*

*users that not all tropical regions are monitored with the same number of IR images, as indicated with the hatched areas. Therefore, the analysis and conclusion have to be carried out with precaution.*

*We have brought some modification to the paragraph to clarify these points:*

*Line 506: "Figure 10 shows the daily 1°x1° spatial distribution of the DCS density calculated from January 2012 to December 2020 over the entire tropical belt. The hatched areas indicate that the southern part of the Western and a southern band between 118° W and 108° W of the Eastern Pacific regions have been affected by missing data respectively from January 2012 to May 2015 and January 2012 to November 2017 respectively. The DCS density of these two specific regions have then been computed with a lower number of days involved. This may impact the patterns of DCS density, and therefore we would like to emphasize to users that the analysis should be carried out with caution. Also, note that the map was built considering only days not impacted by any tracking interruptions over the any part of the tropical region and corresponding to 61.5 % of the total number of days over the entire 2012-2020 period. This number is mainly explained by the interruption of the southern scans of GOES-13 and GOES-15 due to Rapid Scan Operations."*

*We have also corrected the title of the figure from 'Daily MCS density' to 'Daily DCS density'.*

29)      Line 532 – "The CACATOES 1°x1°-1day global monthly files" please clarify what is 1day in this sentence.

*Thank you for your comment. However, the dataset has already been described paragraph 4.3.*

*For more clarity, we have modified global by tropical and changed the sentence as follow:*

*The $1°x1°$-$1day$ CACATOES tropical monthly files, describing the characteristics of DCS overpassing the daily 1° lon/lat grid boxes, are also produced in a NetCDF-4 format with metadata following the Climate and Forecast (CF) Convention version 1.6 and Attribute Convention for Dataset Discovery (ACDD) version 1.3.*

30)      Line 535 – Is the TOOCAN algorithm available? It will be important for the user to have access to the algorithm that produced the dataset.

*The TOOCAN algorithm is not yet publicly available. Our initial focus was on publishing the DCS datasets, and we are considering providing access to the software at a later stage.*

**Citation**: https://doi.org/10.5194/essd-2024-36-RC1

---

## Author Comment (AC3)

**Reviewer #2**

The manuscript introduces two novel datasets about deep convective systems over the intertropical belt: TOOCAN and CACATOES. The TOOCAN dataset contains the tracking of deep convective systems from 2012 to 2020, which is generated using a 3-D tracking algorithm and a homogenized geostationary infrared brightness temperature archive. The CACATOES dataset is derived from the TOOCAN dataset by projecting the morphological characteristics of each deep convective system onto 1-degree x 1-degree grids. Overall, the manuscript is written and organized well. The authors clearly describe the source datasets and explain how to generate the target datasets. Furthermore, the authors discuss the uncertainties of the datasets and briefly compare them with other existing datasets. I recommend the publication of the manuscript if the authors can correct minor language errors.

*We would like to thank the reviewer for the positive comments, appreciations and encouragements. The reviewer #2 has made a number of comments that will benefit this manuscript. We have thoroughly revised our manuscript and addressed all the questions and concerns raised by the reviewer #1. Detailed responses to their comments are provided in the following. the original comments of the reviewers are in black and our responses in blue and italic.*

*We have also adjusted the manuscript accordingly to the journal standards:*

- *The reference list has been compiled according to the journal standards.*
- *We have adjusted the color schemes in the figures 1 and 2 by using a Color Universal Design (CUD) palette, to ensure that colors are distinguishable by everyone, combined with some Different line styles and markers to enhance distinguishability. All the figures have been checked using the Coblis–Color Blindness Simulator to check for distinguishability. Figs. 1, 2, 7 have passed the test. More specifically, we have determined that Fig. 7 is sufficiently distinguishable for individuals with color blindness without any modifications.*
- *Finally, the color cells of the Tables have been removed to fit with the journal requirements.*

*As supplement to the paper, a video (https://doi.org/10.5446/68200, Fiolleau 2024) has been added to the paper to illustrate at a 30-minute time frequency, showing the DCS identified by TOOCAN in October 2015 over the Western Pacific region.*

Minor comments

1) Line 40: "the various the phases" to "the various phases"? "its evolution is" to "its evolution in"?

*Thank you for your comment. We corrected accordingly.*

*Line 39: Despite a long research history (Houze, 2018), understanding the lifetime duration of tropical convective systems and the lengths of their various phases throughout their life cycles in the current climate, as well as their evolution in a warmer and moister world, remains challenging (Roca et al., 2020).*

2) Line 47: Are all DCS events initiated from several individual deep convective cells? Why can't deep convection be initiated from a single deep convective cell?

*Thank you for your comment.*

*You are right. deep convection be initiated from a single deep convective cell as well as several convective cells. We have corrected the sentence in the text:*

*Line 48: "DCS initiate and develop from one or more individual deep convective cells"*

3) Line 53: "infirm"? Do you mean "weaken"?

*We have replaced by "refined".*

4) Line 90: "then" to "the"?

*Done*

5) Line 97: Correct the citation format of Endlich and Wolf 1981.

*Done*

6) Line 110: "multi-thresolding" to "multi-thresholding"?

*Done*

7) Lines 117-120: Please rewrite this sentence.

*Thank you for your comment*

*We have corrected the sentence as follow:*

*Line 117: "These climatologies provided an initial perspective at tropical scale, revealing the ubiquity of mesoscale systems with various durations and spatial extents across a wide spectrum of large-scale environments. This insight has prompted numerous scientific investigations."*

8) Line 182: "zenithal" to "zenith".

*Done*

9) Line 186: Delete "a"?

*Done*

10)    Lines 223-224: Delete "making the processes easier"?

*Done*

11)      Line 237: "the all" to "all the".

*Done*

12)      Lines 281-282: Please rewrite the sentence.

*Thank you for your remark:*

*We have reworded the sentence:*

*Line 285: "Thanks to its spatio-temporal region-growing technique, the TOOCAN algorithm can track DCS by suppressing split and merge artifacts throughout their life cycles, which are inherent to classic overlap-based tracking techniques."*

13)      Line 344: Delete "a"

*Done*

14)      Lines 362-365: If consecutive images are deleted during other hours, how about the results?

[Figure]

*Figure: Diurnal cycle of the initiation of DCS occurence over the 40°W-40°E; 30°S-30°N region in June-September 2012.*

*The figure shows the diurnal cycle of DCS initiation over the 40°W-40°E; 30°S-30°N region from June to September 2012, revealing that the majority of DCS initiate predominantly in the afternoon. It clearly indicates that having a gap of successive missing data in the afternoon is much more impactful than having it between midnight and 5 AM.*

*Actually, as discussed in the response tp the Reviewer #1, Several scenarios arise for the DCS facing such a time period of consecutive missing images. DCS that were supposed to initiate during this missing data gap are either detected at the time of resumption, in which case their duration is artificially shortened, or they cannot be identified at all. DCS that were supposed to dissipate during this period of missing images could dissipate at the time of resumption, in which case their lifetime duration is artificially extended. There is the case of DCS that started before the series of missing images and are expected to persist thereafter can still be tracked despite this interruption. Some DCS cannot survive the period of missing data and then dissipate artificially. Smaller systems may not survive this interruption, reducing their lifetime, or their lifetime may be artificially extended due to data replication.*

*The maximum period of a 3-hour consecutive missing image interval was mainly chosen to ensure continuity in tracking large systems, particularly during eclipse periods affecting METEOSAT-7 observations in India, which occur in the evenings between August and September, as well as between February and March. The bias for long-lived systems (Figure 7) suggests that DCS can still be monitored despite the interruption if they possess sufficiently extensive cloud cover and if the convective situation evolves minimally during the gap.*

*This is now conveyed in the 3.2.b section in a more detailed form.*

> 15)     Line 437: What do you mean by the total cold cloudiness? Cold cloudiness area?

*With total cold cloudiness, we aimed to refer to the geographic area covered by clouds with temperatures below a certain threshold (e.g., 235K).*

*However, you are correct. "Cold cloudiness area" appears to be a more dedicated term. We have replaced 'total cold cloudiness' with 'cold cloudiness area' in the text.*

> 16)     Line 453: "Similarly, to" to "Similar to".

*Done*

> 17)     Line 455: Add "in" before "Fig. 9c".

*Done*

> 18)     Lines 481: Correct the sentence!

*Done*

*Line 497: Roca and Fiolleau (2020) have shown that a couple of DCS impacts significantly each daily 1°x1° grid box, while most of the other systems have very small contributions to the cold cloudiness.*

**Citation**: https://doi.org/10.5194/essd-2024-36-RC2

---

## Author Response (AR2)

*Dear Editor,*

*Thank you for your comment and for providing us the opportunity to clarify the technical points on the quality flag. Below, you will find the original comments from the reviewers in black, followed by our responses in blue and italic.*

**Public justification (visible to the public if the article is accepted and published)**: I would like to thank the authors for the revised submission that addresses the reviewers' comments.

Before publication, I would like the authors to provide a bit more detail on one of the reviewer comments, which I still find a bit confusing. Using plain language rather than referring to a flag that merges several different criteria, would be good. For example, 10xxxx and 01xxxx have very different numeric values, but in the end one has missing images at the end and the other at the beginning. It is not clear to me that these numbers are quantitatively comparable.

Line 407 – ". A quality flag lower than 11110 is a good compromise to keep DCS little impacted by imagery issues". > I feel that this section should still be reformulated given that the quality flag, which I assume is binary/ bitwise merges several different qualities. Could you please specify here what requirements have to be met to be included. I suggest to use plain language rather than the quality flag code.

*Thank you. We have improved the paragraph for clarity.*

*Line 419: Quality control can raise a number of issues with the data, such as missing lines or images which can affect the tracking of systems in various ways. The user is informed of the issues thanks to a flag coded as a five digits number. The first digit indicates whether a DCS is born naturally or due to consecutive missing images, while the second digit indicates whether a DCS dissipates naturally or as a result of consecutive missing images. The third digit reveals if a DCS is impacted by the edges of the image, some missing lines or pixels. Values of these three digits greater than one indicate problems in initiation, dissipation, missing lines or touching the edge of the image. The last two digits of the quality flag represent the number of missing images during the lifecycle of a given DCS. A typical conservative filtering approach would include only systems unaffected by quality control issues, identified by a flag value less than 11110.*